# Expanding *N*-glycopeptide identifications by modeling fragmentation, elution, and glycome connectivity

Joshua Klein [1] ✉, Luis Carvalho [1,2] & Joseph Zaia [1,3] ✉

Accurate glycopeptide identification in mass spectrometry-based glycoproteomics is a challenging problem at scale. Recent innovation has been made in increasing the scope and accuracy of glycopeptide identifications, with more precise uncertainty estimates for each part of the structure. We present a dynamically adapting relative retention time model for detecting and correcting ambiguous glycan assignments that are difficult to detect from fragmentation alone, a layered approach to glycopeptide fragmentation modeling that improves *N*-glycopeptide identification in samples without compromising identification quality, and a site-specific method to increase the depth of the glycoproteome confidently identifiable even further. We demonstrate our techniques on a set of previously published datasets, showing the performance gains at each stage of optimization. These techniques are provided in the open-source glycomics and glycoproteomics platform GlycReSoft available at https://github.com/mobiusklein/glycresoft.

Protein glycosylation is the most heterogeneous PTM[1–3], with effects on a wide array of biological processes[1]. Mass spectrometry has been established as one of the best tools for high throughput analysis of the glycoproteome[4]. However, intact glycopeptide MS/MS interpretation is a challenging problem to address as data are generated with a variety of dissociation strategies and energies, depending upon their appropriateness for different types of glycopeptides in glycoproteomics large and small[5–8].

Depending upon the characteristics of LC-MS/MS used, different software strategies are needed[9]. As many have argued, high-confidence glycopeptide identification requires confident identification of the peptide and the glycan independently[10–12]. SCE collisional dissociation has been shown to be ideal for acquiring more complete fragmentation of *N*-glycosylated glycopeptides[5,8,10]. pGlyco3[7] has been designed specifically with these characteristics in mind, including a GPSM scoring model and multi-dimensional FDR estimation procedure for controlling the peptide and glycan FDR jointly and independently. pGlyco3 uses a large database of glycan structures compiled from GlycomeDB[13] and other sources, to be able to exactly enumerate glycan fragments for a coverage calculation central to their method.

Previously, we presented a method for quality controlling in glycan identifications using an RT model[14] that could be applied post-search to determine if the assigned glycans were consistent or whether they may be isobaric substituted ammonium adducts or sulfated. More recently, pGlyco3, GlycoDecipher[15], and MSFragger-Glyco[16] presented search engines that could identify adducts but did not attempt to disambiguate them. We present an extension of our previous method for RT modeling that can correct the misidentification of glycan compositions with specific isobaric substitutions without needing prior knowledge of the chromatographic column or the gradient schedule used, provided at least two glycoforms are identified for a peptide backbone and an uncertainty measure for each glycopeptide meeting those requirements.

Collisionally dissociated glycopeptide MS/MS spectra can be complex, having peptide b and y ions with or without glycan reducing end residues, glycan B ions, and intact peptide + glycan Y ions, with varying abundances in varying charge states depending upon the number and strength of the bonds in the precursor molecule and the collision energy used[6]. Thus far, only glycan B ions have received critical analysis about their inter-relationships[17,18], and limited work has been done on broad

[1]Program for Bioinformatics, Boston University, Boston, MA, US. [2]Department of Math and Statistics, Boston University, Boston, MA, US. [3]Department of Biochemistry and Cell Biology, Boston University, Boston, MA, US. ✉e-mail: joshua.adam.klein@gmail.com; jzaia@bu.edu

glycopeptide spectrum prediction[19] while these topics have been well explored and exploited in peptide spectrum matching[20–25].

We present a collection of methods to learn inter-peak relationships based on their fragment ion annotations and to learn to predict the relative intensity of glycopeptide fragmentation events across a wide range of charge states. In addition, we present a generalization of our glycan network smoothing technique[26] to construct models of site-specific glycosylation to guide glycopeptide identification spanning the modeled sites. We provide an implementation of these techniques with GlycReSoft and its supporting libraries.

## Results

### Workflow of GlycReSoft

GlycReSoft is an end-to-end search space builder, MS2 identification, and MS1 feature extraction tool for glycoproteomics with adducts or mass shifts. It builds databases of glycoproteomics from FASTA, PEFF[27], or mzIdentML[28], and glycan compositions from combinatorial expansion, biosynthetic simulation, or a user-provided list as a text file in a composable dialect of IUPAC[29] (Fig. 1a). When available, it queries UniProt[30] for additional annotations and those in PEFF to add non-enzymatic cleavage sites during protein digestion. Prior to searching a sample, GlycReSoft applies LC-MS/MS deisotoping and charge deconvolution to produce known charge state neutral mass peak lists and correct precursor monoisotopic masses (Fig. 1b).

It uses a glycan-first search strategy[7] to filter the space of glycopeptides for each mass shift (Fig. 1c), and scoring of the top $k = 70$ glycan mass-peptide mass combinations per mass shift proposed for each spectrum (Fig. 1d). Following searching all spectra, it estimates false discovery rates (FDRs) for the peptide and glycan components (Fig. 1e). After retaining only glycopeptide spectrum matches (GPSMs) that pass the required minimum joint FDR threshold, GlycReSoft extracts MS1 features, maps retention time (RT) spanning, precursor mass matching GPSMs to those features, and assigns the glycopeptide with the greatest sum of MS2 scores to each MS1 feature (Fig. 1f).

Recently, we and others have increasingly looked for modified or adducted glycans[7,14–16], including those that are difficult to discriminate from MS2 spectra alone like ammonium adduction. GlycReSoft re-assigns glycopeptides of MS1 features when they are assigned with a mass shift and do not have an overlapping un-shifted version (Fig. 1g). Then GlycReSoft iteratively fits a relative RT model from glycopeptides with and without mass shifts, later with all glycopeptides similar to those already covered by the model (Fig. 1h). After each iteration, it uses a set of glycan composition and mass shift revision rules to update identity assignments based upon their observed RT versus the predicted RT for each alternative glycan composition (Fig. 1i). Finally, it assigns each glycopeptide that the RT model covers an RT score. For more information, see supplementary Section 11.1. Sometimes, GlycReSoft is unable to determine a single best solution and reports multiple configurations of the same peptide for the same spectrum with differing scores. When discussing the number of spectrum matches produced here, we refer to only each spectrum exactly once, using the configuration that best matches the chromatographic feature if possible.

GlycReSoft's MS2 scoring model can be enhanced by learning to predict fragmentation patterns (Fig. 1j) which may be instrument or dissociation method-specific[21,24]. In addition to modifying the peptide and glycan scores themselves, the peptide correlation is used as a feature during peptide FDR estimation. It can also learn biologically meaningful relationships about the kinds of glycans that appear at a particular glycosylation site for a particular glycoprotein (Fig. 1k) extending previous work[26].

### Comparison with pGlyco3 on mouse tissue *N*-glycoproteomics

We applied pGlyco3[7], the base GlycReSoft algorithm, and the extended scoring models learned from those initial GlycReSoft results to the five mouse tissue datasets composed of five replicates each originally published in ref. 10 (PXD005411, PXD005412, PXD005413, PXD005553, and PXD005555) with the total number of GPSMs identified at 1% FDR shown in Fig. 2a. GlycReSoft's base method identifies 6.2% more spectra than pGlyco3, though it does marginally worse on the brain tissue dataset, identifying 3.6% fewer spectra. On average pGlyco3's glycan FDR threshold is lower, requiring fewer fragments to produce acceptable matches, due in part to its exact enumeration of glycan fragments and differing coverage weights. We exported annotated MGFs of identified spectra exceeding a total score of 20 for each sample and for each tissue, type fit glycosite-specific glycan network smoothing models for glycopeptides identified, passing a 1% joint FDR threshold. For a complete overview of the overlap between pGlyco3 and each scoring model in GlycReSoft see supplementary Figs. 8, 9 and 10. In addition, see supplementary Fig. 11 for a comparison of glycan databases between GlycReSoft and pGlyco3.

To compare the influence of retention time modeling, we re-scored GPSMs from the base GlycReSoft identification method and pGlyco3 for each mouse tissue LC-MS/MS experiment with the RT model for that experiment, retaining the highest RT scoring GPSM for each glycopeptide in it, omitting any glycopeptide not covered by the RT model for that experiment. Each run's retention time model covered 72% of glycopeptides identified on average. GlycReSoft reported 39161 glycopeptides with a high-confidence RT score over 0.75 (defined in supplementary Eq. S12, loosely approximating the probability of matching a time point by random chance of ≤ 25%), compared to pGlyco3's 35687, 9.7% more, and only 2427 glycopeptides with a low confidence RT score of 0.10 or lower compared to pGlyco3's 5497, 44% fewer shown in Fig. 2b. We show the coefficients of the monosaccharide features over time in Fig. 2d whose stability is influenced by the number of training data points at that time in the upper plot and the vertical lines denote the boundaries of overlapping model bins which tend to fall at bin centroids, for all other samples in the mouse tissue dataset, see supplementary section 10.4. The large number of mass shifted GPSMs supported by a retention time model score of at least 0.2 suggests that a tool that does not consider retention time overestimates the abundance of unmodified or adducted glycopeptides when it is favorable to the scoring function to perform such a substitution, shown in Fig. 2e.

Because the primary substitution in the brain tissue dataset swaps d-Hex and NeuAc, edge cases may be detected by the presence or absence of diagnostic ions which influence match selection early in the search process. The middle ground where the addition of another d-Hex and removal of a NeuAc from a composition, or the inverse, without completely eliminating one of those monosaccharides from the composition is not easily resolved by diagnostic ions alone. For the brain subset where both pGlyco3 and GlycReSoft's base algorithm identified the same spectra and the retention time model covered the glycopeptide, we observed that for glycans pGlyco3 identified with {d-Hex:1, NeuAc:1}, GlycReSoft found 56% of the glycans reported {d-Hex:2} with an adduct. Likewise, for {d-Hex:2, NeuAc:2}, GlycReSoft reported 83% {d-Hex:3; NeuAc:1}. In both cases, adding more d-Hex to the glycan composition makes the glycan composition harder to cover, as well as more difficult to disambiguate due to the presence of existing diagnostic ion-contributing monosaccharides. This illustrates the importance of RT modeling for accurate monosaccharide assignments.

We evaluated the fragmentation modeling method's robustness by using a model fit on all but the brain tissue samples and showed that GlycReSoft with the fragmentation model performs 5.7% better on brain samples compared to pGlyco3, but in total 13.4% better overall, shown in Fig. 2a. We show the differences in the model's ability to predict the intensity patterns for peptide backbone fragments and peptide+Y ion fragments in Fig. 2f and g respectively, with an overall median correlation of 0.833 on the training spectra and 0.801 on the

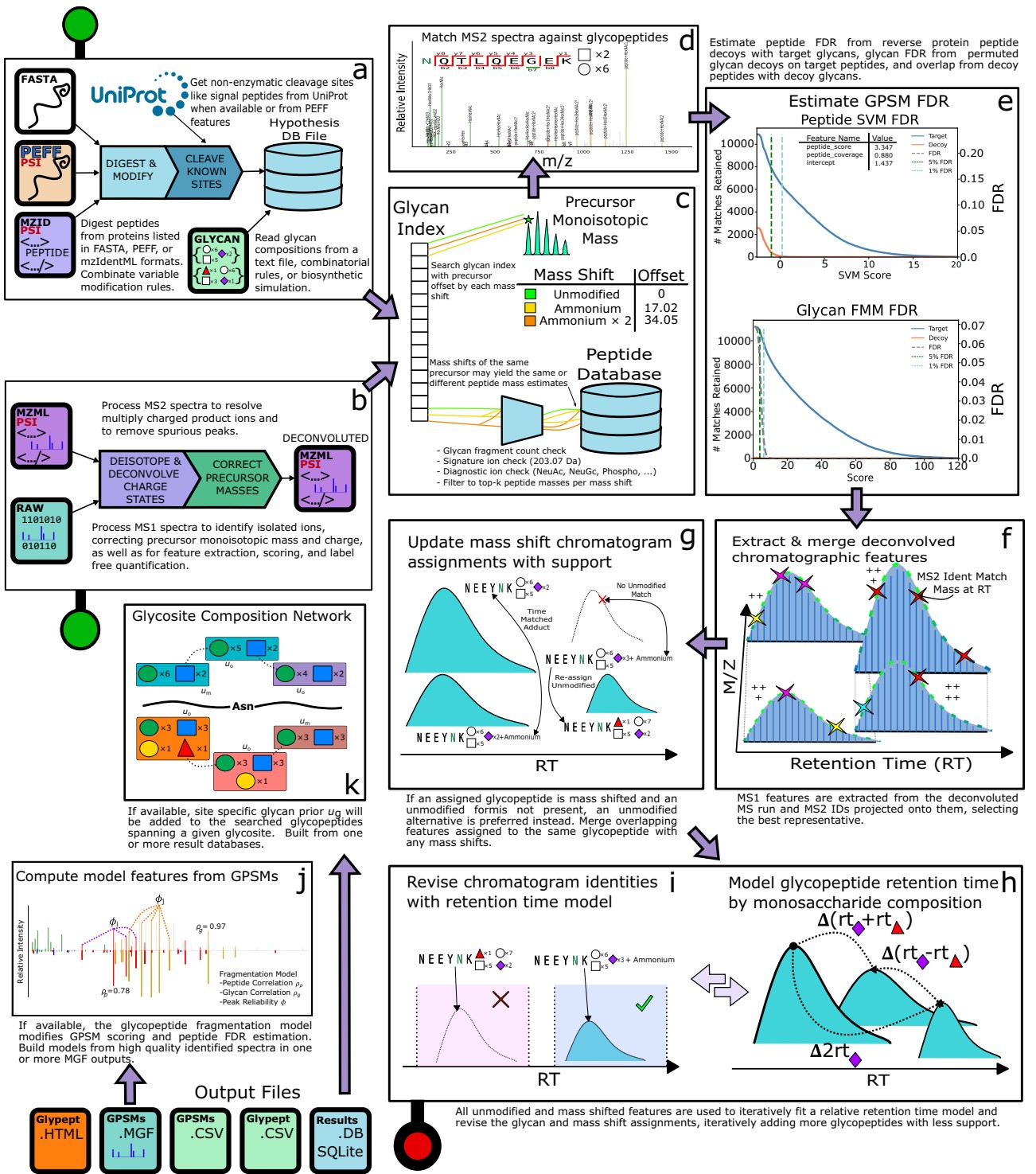

**Fig. 1 | An overview of GlycReSoft.** GlycReSoft's workflow stages. **a** Search space construction, **b** LC-MS/MS deisotoping and charge deconvolution, **c** Scanning the complement glycan ion index for each mass shift, **d** Full spectrum match re-scoring, **e** FDR estimation, **f** MS1 feature extraction and MS2 identification mapping, **g** Parsimonious updates to glycopeptide-mass shift-MS1 feature assignment, **h** Relative retention time model, **i** Glycopeptide revision based upon retention time, **j** GPSM feature scoring from fragmentation model if available, **k** Glycosite glycan network smoothing score adjustment if available.

test spectra, broken down by charge state in supplementary Fig. 6. The difference between the training and testing tissues were relatively close, within 3% for both the peptide and glycan component, example predicted spectra are shown in supplementary Figs. 14, 15 for train and test respectively. The peptide correlation plays a direct role in the peptide FDR, being used as an SVM feature, as well as a component of the peptide score (defined in Eq. (12)). The addition of fragmentation modeling increases the median number of GPSMs passing 1% peptide FDR by 4.5% (0.5% to 9.5%). The median number of identifications passing the 1% glycan FDR increased by 2.5% (−1.6% to 12%) while using the same estimation method as the base search strategy on the model-augmented glycan score (defined in Eq. (17)).

We built upon our previous work in biosynthetic network-based glycomics modeling from Klein et al.[26], which modeled the glycome of

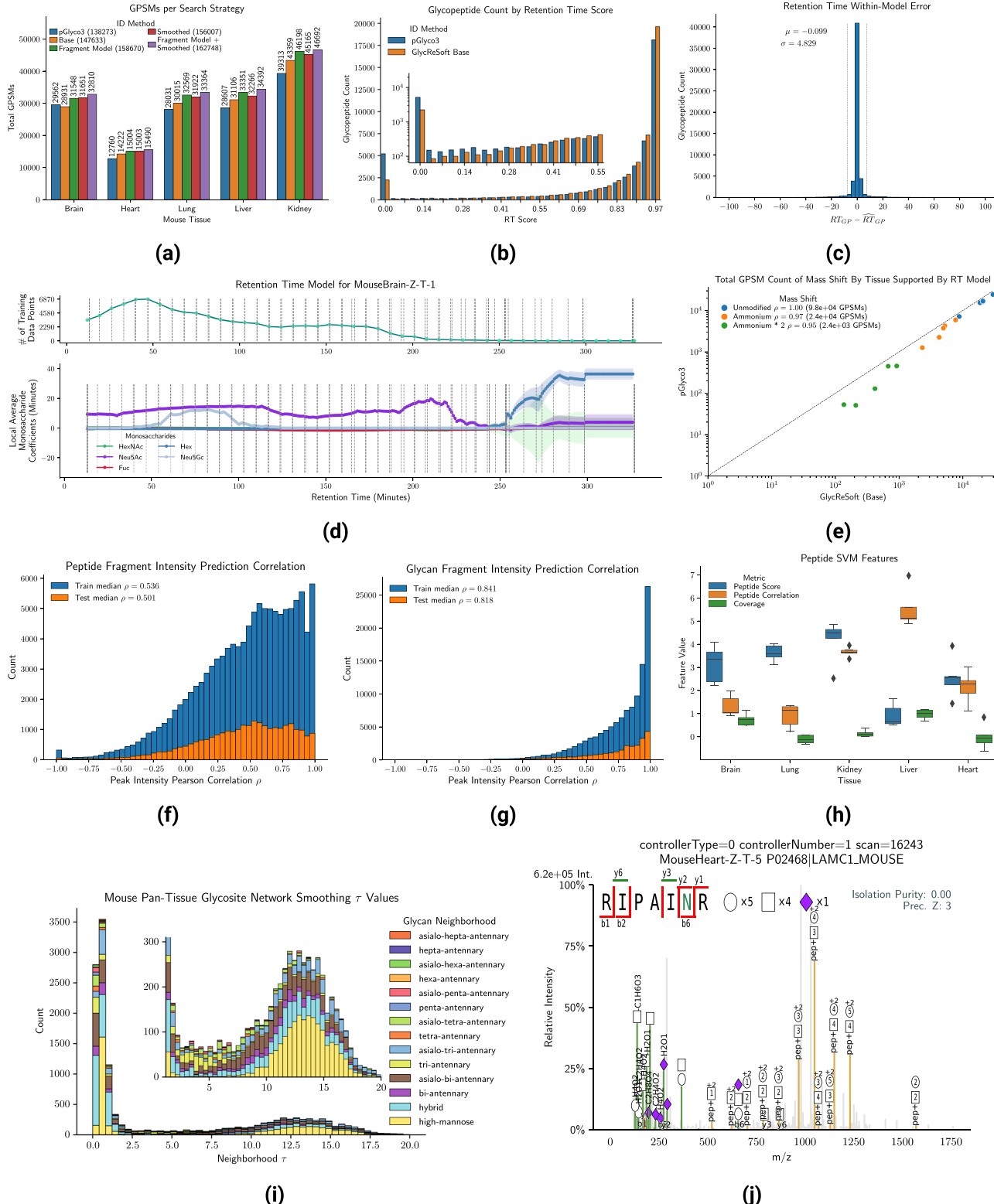

**Fig. 2 | Impact of modeling glycopeptide identification elements for the mouse tissue datasets. a** The GPSM counts for each search strategy on each mouse tissue dataset at 1% FDR, **b** The total GPSM counts across runs for each retention time score, **c** The distribution of retention time residuals in minutes, **d** The retention time model's local training data and monosaccharide coefficients over time with approximated confidence intervals for each monosaccharide over time derived from a variable number of observations totaling in the upper line plot, **e** The number of GPSM counts by mass shift type for each tissue with a minimum retention time model score of 0.2, **f** The peptide Pearson correlation for the training tissues (Heart, Kidney, Lung, Liver) and test tissue (Brain) listing the median value in the legend, **g** The glycan Pearson correlation broken down as the peptide correlation, **h** The median/quartiles of peptide FDR SVM model feature weights by tissue when fragmentation model was used estimated from the five replicates for each tissue, **i** The network smoothing parameter means $\tau$ for each glycosite across all mouse tissues, **j** A GPSM rescued by network smoothing that only passed the 1% FDR threshold after network smoothing. Source data are provided as a Source Data file.

a released glycan glycomics experiment in the aggregate to instead model the glycome of each glycosylation site we detect glycopeptides for in the glycoproteome. We estimated glycan network smoothing parameters for each glycosite observed in each tissue type across each set of five replicates where there was an identified glycopeptide with a total FDR below 0.01. Briefly, this method estimates some glycan biosynthetic neighborhood central tendency parameters $\tau$ based upon observed glycan evidence, a graph describing the biosynthetic pathways relating glycans to each other, and a given weight matrix $\mathbf{A}$ which expresses how much weight a neighborhood has on each glycan, and a smoothing parameter $\lambda$ controlling how much influence each glycan has on $\tau$ which is selected using grid search. $\tau$ is analogous to the weight each neighborhood in the biosynthetic graph has at a given site, and $\mathbf{A}$ maps those weights onto each glycan in the search space. Once we have a point estimate of $\tau$, we can use it, $\mathbf{A}$, and $\lambda$ to infer some likelihood $u_g$ that an unobserved glycan composition implied by the observed glycan compositions is also present, but with insufficient evidence to be confidently identifiable alone, or to increase our confidence in observed glycans at the same site from the same neighborhoods. We limited the maximum value of the smoothing parameter $\lambda$ to 0.2 to prevent over-smoothing when estimating $u_g$. We repeated each search using the glycosite network smoothing models for each tissue, which modifies the glycan score (defined in Eq. (19)). The non-zero site-specific $\tau$ estimated for the five tissue datasets are shown in Fig. 2i, showing strong signal for high mannose and small hybrid or complex-type $N$-glycans, though these neighborhoods are also the most concentrated and fastest growing, while larger neighborhoods' parameters are distributed more broadly. Because most glycosites are observed with only a small number of distinct glycoforms, those larger neighborhoods tend to have a large number of glycosites where their parameter is at or near zero, as shown in supplementary Fig. 42. When searching decoys, decoy proteins received the same site models as their target counterparts, with site associations preserved after sequence reversal and decoy glycans were treated identically to their target counterparts, receiving the same value of $u_g$ ensuring that targets and decoys are not treated differently.

GlycReSoft with site-specific network smoothing retained 4.9% more GPSMs than the base scoring model, shown in Fig. 2a. Network smoothing helps glycopeptides which came from glycosites that were modeled as well as glycopeptides whose confidence estimates were on the border of passing the required FDR threshold. An example spectrum match is shown in Fig. 2j which was rescued by network smoothing, with the broader glycosite coverage shown in supplementary Fig. 12. On average, target-peptide glycopeptides that came from smoothed glycosites had glycan prior contributions $u_g$ 222% better than decoy peptide-glycopeptide, though this only translated to a overall 2% score advantage for targets compared to the base scoring model shown in Fig. 3a. This bias altered the FDR landscape, at 1% FDR smoothing increased the number of GPSMs retained by 3% on average for the peptide and by 4.5% on average for the glycan component. We repeated the entrapment FDR experiment from[10] using the fission yeast glycoproteomics dataset from PXD005565, concatenating the fission yeast and mouse reference proteomes and glycomes and compared the ratio yeast peptides with yeast glycans to all other combinations of glycopeptides, using GlycReSoft and GlycReSoft with network smoothing enabled respectively, using the model learned from mouse brain, and pGlyco3, all considering up to two ammonium adducts. The glycan databases were equivalent to the mouse tissue dataset, just with the addition of yeast glycans to the GlycReSoft database, while the pGlyco3 database already contained them. We found that using the base algorithm, GlycReSoft had an entrapment FDR of 2.84%, GlycReSoft with network smoothing had 2.85%, and pGlyco3 had 1.95% in Fig. 3b, showing that the network smoothing model had a trivial impact on entrapment. While GlycReSoft does produce more mouse glycan identifications, the primary source of its elevated entrapment FDR relative to pGlyco3 is from mouse peptide identifications from spectra pGlyco3 did not assign anything to.

We re-estimated the glycan network smoothing parameters from the search results in section 2.2 and repeated the searches. The combined method identified 9.5% more GPSMs compared to the base method, as shown in Fig 2a, with 2.2% more than fragmentation modeling alone and 4.5% more than network smoothing alone, a gain of 15.7% overall compared to pGlyco3.

### Comparing across datasets and collision energies

We applied GlycReSoft to samples from two different studies acquired on the same instrument model, a Q-Exactive (Thermo, San Jose, CA), but at different collision energies PXD005931 of 24 human serum samples screened for prostate cancer[31] at 15/25/35 nCE and PXD009654 of 6 human serum samples, half hepatocellular carcinoma and half normal,[32] at 20/30/40 nCE. Both studies enriched for sialylated glycopeptides from the human proteome, substantially shrinking the complexity of the glycoproteome. We fit a fragmentation model on PXD005931 and tested it on PXD009654 and compared the performance on spectrum prediction and database search performance. With the exception of one sample, both models yielded a modest improvement in GPSMs at 1% FDR, + 1.0% for PXD005931 and + 4.3% for PXD009654. The model trained on PXD009654 performed substantially better, shown in Fig. 3d since this is akin to training on the test set. Both models had useful correlation predictions, 78.5% vs. 92.5% glycan correlation and 56.9% vs. 69.2% peptide correlation respectively. The median total glycan reliability $\sum\phi_G$s were 7.0 and 7.1 and the median total peptide reliability $\sum\phi_P$ were 4.5 and 4.8, respectively. When we searched PXD005931 with its own fragmentation model, we observed a similarly modest improvement in the depth of identifications, + 3.5%. The complete breakdown of GPSMs counts by strategy is shown in supplementary Fig. 1.

In sample HCC_2 from PXD009654, we found that the fragmentation model caused a large decrease in GPSMs. This was caused by one decoy glycan match receiving a marginally better score. That this sample identified 14% more spectra than the others in the same condition when searched using the base algorithm may be spurious. When this sample is omitted, the percent improvement in GPSMs at 1% FDR is + 3.1% for the PXD005931 fragmentation model and + 6.5% for the PXD009654 fragmentation model. In both datasets, the peptide FDR appears to be the limiting factor to a much greater degree than in the mouse tissue dataset. The ratio of GPSMs $\frac{\#\text{passing glycan FDR}}{\#\text{passing peptide FDR}}$ has a mean of 2.1 for the mouse dataset, 2.6 for PXD005931, and 3.5 for PXD009654.

To test whether it was the dataset or the model that produced diminishing returns, we applied the fragmentation model trained on PXD005931 to the mouse tissue dataset which used a different type of mass spectrometer. The glycan intensity correlation was 0.56, and the peptide intensity correlation was 0.19, substantially lower than the model trained on the mouse dataset. Both models' median total reliabilities were very close, median total glycan reliability $\sum\phi_G$s were 12.9 for both the PXD005931 and mouse dataset model, and the median total peptide reliability $\sum\phi_P$s were 11.8 and 12.1 respectively. The PXD005931 search results yielded a 2.6% improvement in GPSMs at 1% FDR compared to the gain of 6.5% from the mouse model, shown in Fig. 3e. This difference is observed on both the peptide and glycan estimates, though disproportionately worse for the peptide. Using the held-out brain dataset, we observed a difference of 2.5% between the two models, compared to the much larger difference of 4.1%, suggesting that the difference explainable using the PXD005931 model in a generalizable context is smaller. In addition, given that the $\sum\phi_P$ values for the two models are similar, they are a more plausible explanation for the performance advantage.

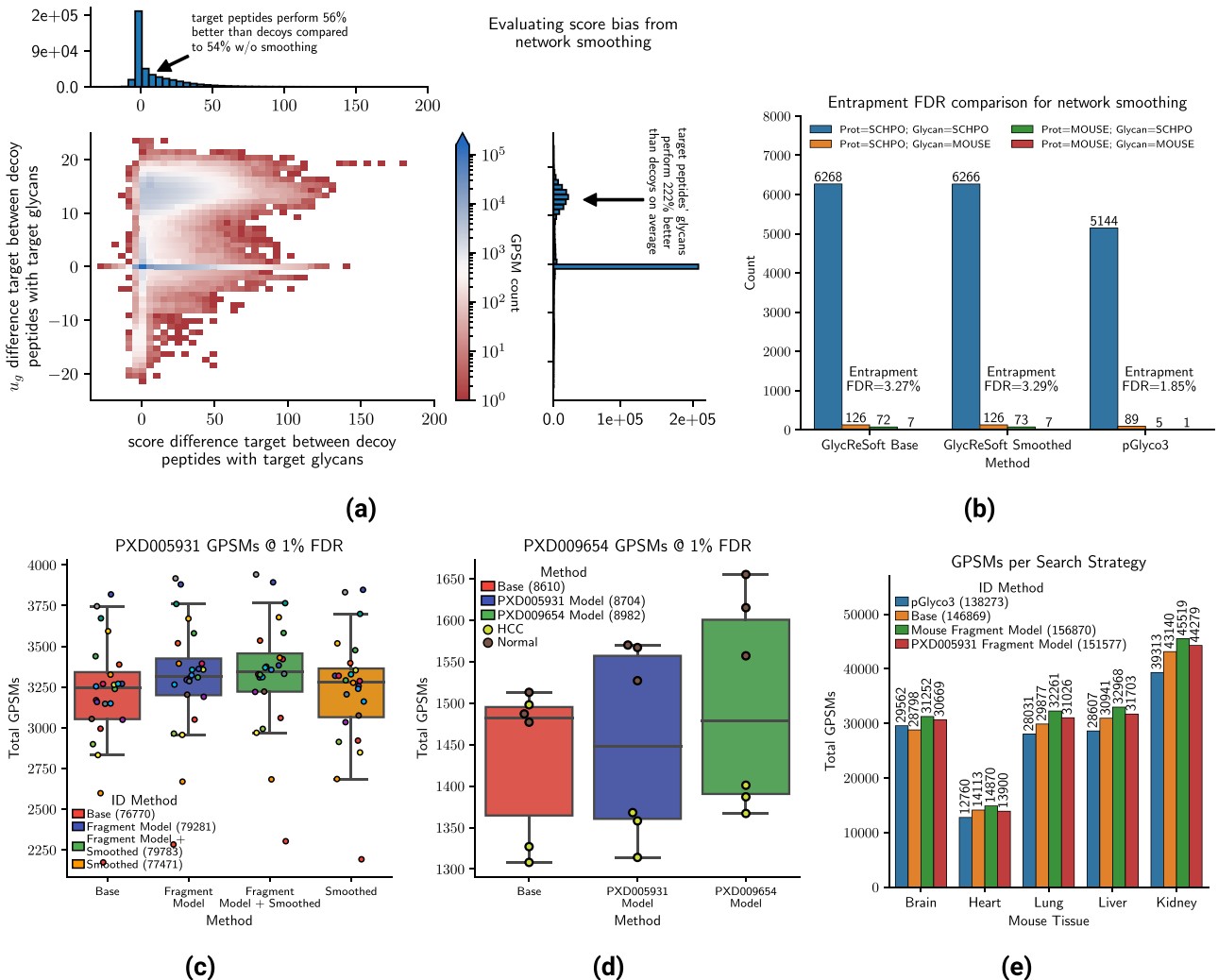

**Fig. 3 | Evaluation of modeling approach on orthogonal controls and other datasets. a** The effect of network smoothing on target- and decoy-peptide pairs for individual spectra showing differences in score and differences in glycan smoothing term $u_g$ for the mouse tissue dataset, **b** Validation of network smoothing with entrapment FDR at a regular GPSM 1% FDR showing it does not appreciably change compared to the base search though see supplementary section 9 for more details, **c** The GPSM counts by search strategy in the human serum dataset PXD005931 at 1% FDR with individual replicates shown as swarm points colored across strategies, with median line and error bars reflecting inter-quartile range of GPSM count within each search strategy from the 24 serum samples, **d** The GPSM counts for the human serum dataset PXD009654 using either the base scoring method, the fragmentation model trained on PXD005931, or the fragmentation model trained on PXD009654 to measure transferability within instrument/species at different collision energies, with median line and error bars reflecting the inter-quartile range over the three replicates per condition, **e** The GPSM counts at 1% FDR for the mouse tissue datasets to compare the transferability of the fragmentation model trained on PXD005931 across instruments, species, and collision energies. Source data are provided as a Source Data file.

## Discussion

The work presented here demonstrates that it is possible to expand the number of high-confidence glycopeptide identifications by up to 15.7% without sacrificing stringency by incorporating physio-chemical, biosynthetic factors, and prior knowledge into the identification process. In addition, we introduced a method for using retention time to detect and correct for a sub-class of quasi-isobaric identifications that cannot be resolved from MS2 spectra alone automatically, a feature not provided by any other method to date.

### Retention time modeling

Our relative retention time model was capable of correcting a large number of errors accurately during the middle of the LC run when a large number of glycoforms were available, but as observed in Fig. 2d, as the number of data points to fit on tapered off, the local model's parameters fluctuated erratically. While it would be possible to prevent extreme fluctuations by applying a weight based on the number of

data points used to train a local model when calculating average retention time (supplementary Eq. S8), this limited the model's unsupervised performance on segmented gradient methods as was done in PXD009654[32], and biased against both the start and end of an LC gradient regardless due to the smaller number of observations. In principle, pooling multiple runs' LC features and modeling them simultaneously would help, but it also introduces substantial technical complexity requiring some level of LC feature alignment and modeling run-level parameters as well as using a shared FDR model during revision testing. While GlycReSoft was written to process individual LC-MS/MS runs independently and leave integration for downstream analyses, this would be a useful future direction. In addition, this method is only appropriate for reverse-phase chromatography with polar column chemistry, and likely would not work for other types of chromatographic columns like porous graphitic carbon[33] or separation techniques such as capillary electrophoresis[34]. Another major limitation of this technique is the need for two glycoforms of a peptide to

even have a chance to be considered, due to the need to have some point of reference. The requirement would be removed if the deglycosylated form were included in the sample as done in SRRCalc's glycoproteomics extension[35] or if an in-silico prediction for the deglycosylated form were available, though these add their own complications. Adding deglycosylated peptides requires additional sample preparation steps and either adjusting the software to identify them as well or a separate deglycosylated peptide-only run to identify chromatographic apex times and modifying the software to receive that information as input. Predicted peptide retention time requires a similar modification to the model, but also requires either a chromatography column-specific model like DeepLC[36] or spiking in iRT peptides to convert iRT predictions to real RT coordinates for a chromatography column-independent predictor like Prosit[25].

GlycReSoft will sometimes report more than one glycan composition for the same spectrum because the retention time model invalidated one explanation, but could not find another explanation that was markedly better than all the others remaining. It still applies parsimony across spectra when determining which glycopeptide(s) best explain a chromatographic feature, though this may conceal poorly separated analytes or co-eluting features that require more information to disentangle. For this reason, we do not yet recommend using the relative RT model to identify arbitrary MS1 features without MS2 spectra in a complex sample, having too many theoretical explanations.

### Fragmentation modeling

The fragmentation model we proposed borrows from two sources, one for increasing the weight of complementary fragmentation events, and another for preferring solutions where the expected product ion abundance matches the observed abundance. While our intensity correlation is not comparable to those results shown on regular tryptic peptides[23-25], we still demonstrate its utility in Fig. 2h, being more valuable than the peptide backbone coverage alone when separating targets from decoys for every sample in the mouse study. We compared our multinomial logistic regression model with gradient boosting regression using the `scikit-learn` library and a parameterization similar to MS$^2$PIP[24,37] and found that while the gradient boosting tree performed slightly better on the held-out brain dataset with a median $\rho = 0.548$ compared to $0.501$, it came at the cost of substantially more over-fitting with a median $\rho = 0.814$ on the other mouse tissues shown in supplementary Fig. 2. This may be due to the lack of diverse peptide backbones found in the mouse tissue dataset, given the limited sequence space surrounding N-glycosites.

Fang et al.[15] demonstrated how the peptide backbone fragmentation patterns of different glycoforms are strongly correlated. We measured the correlation of b and y ion intensities within the set of training spectra for the mouse tissue dataset grouping according to peptide backbone and precursor charge, finding a median $\rho = 0.91$, suggesting that any form of library building or multi-round search will perform better than our peptide backbone fragment intensity predictor. A similar strategy was used by Yang et al.[38] in GproDIA for peptide+Y ions, where the average of the intensity series for a given glycan attached to three similar peptide sequences was substituted for another sequence, with mass shifted to match the new sequence when that glycoform was not found in the spectral library. We applied the same strategy, averaging over all peptide sequences in the mouse training spectra per glycan composition charge state pair, finding $\rho = 0.73$ or $\rho = 0.71$ when using just the average of its three nearest peptide neighbors. This is less than the performance of the model trained on the same data evaluated on the held out test spectra, $\rho = 0.818$ and $\rho = 0.841$ on the training spectra. While our model performed better even on a harder task, it can't be used entirely independent of experimental spectra either, as it requires a hint about whether the intensity series is ascending or descending as the glycan dissociates, described in the extended methods section 4.6. Our attempts to model which direction the ion ladder would follow based on peptide sequence, charge, and charge mobility were better than random chance but were not consistent across all partitions of the explored datasets. Still, our model, depending upon only peptide backbone amino acid composition, glycan size, and proton mobility generalized so well suggests that this task is amenable to in-silico prediction with models simpler than those used today for regular peptide backbone fragments.

We showed that peak intensity prediction did not transfer well from instrument to instrument or at different collision energies in Fig. 3d and e, even though reliability did. While reliability did improve identification quality, it did not improve FDR separation as well as correlation did. We used a Wald test for each feature to test if it was significantly different from 0 to evaluate whether that feature was meaningful, even in the presence of high dispersion. We found 130 significant peptide features, 59 significant descending intensity partition glycan features, and 45 at $\alpha = 0.01$, and the coefficient values are shown in supplementary Section 4. The peptide-related features are consistent with observations for collisional dissociation from Zubarev et al.[39], while the glycan features are centered on peptide backbone composition, particularly charged residues, in both ascending and descending partitions.

### Glycan network smoothing

The glycan network smoothing method we introduced uses information about identified glycan compositions to boost confidence in related glycan compositions based upon different classes of N-glycans that may occupy a given glycosite, capturing the underlying biosynthetic process. While it does not alter the search space, it makes pre-existing matches supported by a site-specific model stronger, which can, in turn, change the average estimate of uncertainty at a given score. Because $u_g$'s contribution in this implementation is limited by the glycan coverage term, it is less helpful to compositions containing d-Hex, which are harder to cover, and it cannot rescue glycosites for which there is insufficient information to fit the model. It is not as strongly affected by instrument settings provided that peptide+Y ions are still produced, making it more portable than the fragmentation model. A glycosite model may theoretically be created from prior knowledge or database annotations from sources like GlyGen, GlyConnect, and GlyCosmos[40-42], which track site-specific glycan identification from the literature prior to collecting empirical evidence, or to aid with identifications that may not have sufficient signal on their own.

The glycome model should be appropriate to the system being studied. If there is little overlap between the glycome of the sample and the provided model, no benefit is had, but if the glycomes overlap but have different pathways like applying a mammalian glycome to a human or other great ape sample where NeuGc and Gal-$\alpha$-Gal groups[43] are absent could skew results towards incorrect compositions. GlycReSoft includes definitions for human and mammalian N-glycan biosynthetic pathway-based networks, but a user may manually define a network in a text file with an arbitrary topology and neighborhoods to estimate $\tau$ from. If there are no known rules governing the glycome of interest, a fully connected grid of observed glycans with small neighborhoods may work best to incrementally propagate information, but care should still be used when evaluating the results.

While the entrapment study shown in Fig. 3b suggests that network smoothing doesn't introduce extreme biases when applied to completely different samples, it does not prevent similar glycan compositions from being misassigned because of a quirk in how they benefit from different $\tau$ parameters at the same site. Combined evidence from multiple sources is still necessary, such as retention time and tandem mass spectra, or additional dissociation modes to identify the glycan composition unambiguously. It may also result in too few decoy glycan matches to estimate the glycan FDR accurately as decoy matches must be as good or better than the target matches.

The decoy glycan approach introduced in pGlyco[44] applies a random mass shift to all peptide+Y ions except for the intact peptide without any monosaccharide residues attached or to the peptide plus one monosaccharide residue, guaranteeing that decoy glycan matches will always reach a similar baseline score to a target glycan where the glycan is almost entirely eliminated, treating such matches as no better than random. This forces the glycan FDR to require some degree of further matching of the glycan in order to pass a reasonable confidence threshold, but it also creates a bias against matches whose peptide+Y ions are low intensity because of the role intensity plays in the score, and as previously mentioned, some glycopeptides' smallest glycan moiety peptide+Y ions are the highest intensity peaks they produce for that ion ladder which gives those decoy glycan matches a disproportionately higher score despite any other peak matching by random chance. We found that not preserving those base-most ions' mass completely eliminated the glycan FDR's ability to discriminate random matches. This suggests that comparing FDR estimates between tools that do not follow the same rules is unhelpful despite having similar "target" scores.

## Summary

We presented three complementary techniques for increasing the depth, breadth, and accuracy of glycopeptide identifications by LC-MS/MS, demonstrating their impact on multiple SCE datasets. These techniques can be applied independently or in combination. Future work remains to improve the prediction of peptide backbone fragmentation for glycopeptides and to generalize beyond instrument and collision energy-specific SCE datasets. While our implementation is a classical "closed search", these techniques would also be applicable to open search strategies and provide an avenue of future improvement. We provide open-source implementations of these techniques online at https://github.com/mobiusklein/glycresoft.

## Methods

### Datasets

We demonstrated our method on two previously published stepped energy HCD glycopeptide datasets. The first dataset, originally published by[10], was enriched glycopeptides from mouse brain (PXD005411), kidney (PXD005412), heart (PXD005413), liver (PXD005553), and lung (PXD005555) tissues, which we will refer to as the mouse tissue dataset as well as the fission yeast dataset (PXD005565) for the entrapment study, all collected using stepped collision energy of 20/30/40 nCE on an Thermo-Fisher Orbitrap Fusion Tribrid instrument. The second was originally published by ref. 31 and was enriched for sialic acid-containing glycopeptides from human serum (PXD005931) acquired using a Thermo-Fisher Scientific Q Exactive with a stepped collision energy of 15/25/35 nCE[32]. The process we used involved a sequential refinement, but at each stage, we used the same processed MS data, glycoproteome databases, and search parameters.

**LC-MS/MS Preprocessing.** We downloaded raw data files for each dataset from PRIDE[45] converted to mzML using ProteoWizard[46], followed by peak picking, deisotoping and charge state deconvolution using GlycReSoft's preprocessing tool[14]. The preprocessing tool averaged each MS1 scan with the preceding and following MS1 scan, did not apply background reduction, used a glycopeptide averagine (H15.75 C10.93 S0.02 O6.47 N1.65) for MS1 scans and a peptide averagine for MSn scans.

**Database construction.** For the mouse tissue dataset, we used the UniProt reference *Mus musculus* proteome UP000000589[30] and extracted only those from SwissProt. We extracted the glycan list from pGlyco3's large prebuilt mouse *N*-glycan database[7,13] and simplified the entries from structures to compositions and combined it with a

mammalian *N*-glycan biosynthetic simulation[29] allowing NeuAc, NeuGc, and Gal-*α*-Gal terminal groups. We combined this proteome and glycome, allowing one glycosylation per peptide, generating peptides using a trypsin cleavage rule allowing up to two missed cleavages, and applied a constant carbamidomethyl modification at cysteine and variable oxidation modification at methionine. We generated decoy proteins by reversing complete protein sequences but retaining *N*-glycosylation sites at the disrupted sequons without introducing new sites as in ref. 7. For the entrapment study, we used the fission yeast reference *Schizosaccharomyces pombe* proteome UP000002485[30]. For the human serum datasets PXD005931 and PXD009654, we used the UniProt reference *Homo sapiens* proteome UP000005640[30] and a human *N*-glycan biosynthetic simulation[29] allowing only NeuAc terminal groups. For PXD005931, we included a short list of common mucin-type *O*-glycans as well.

**Search strategy.** We followed the mass accuracy settings suggested in each dataset's original publication but found that the number and type of adducts to consider were omitted or incomplete. For the datasets in the mouse tissue dataset and the yeast entrapment dataset, we allowed a 5PPM mass error tolerance for precursor ion matches and a 20PPM mass error tolerance for product ion matches, permitting up to two ammonium adducts. For the PXD005931 samples, we used 10PPM mass error tolerance for both precursor and product ion matches and also allowed one ammonium adduct[14], and for PXD009654 we used a 10PPM precursor mass error tolerance. Our search strategy did not consider chimeric or co-isolating precursors, although when we compared results to pGlyco3, we included their chimeric solutions. GlycReSoft searches with identification methods were all subject to the same adduct deconvolution and retention time modeling procedures as described later for consistency.

**Base scoring model**

We built upon the GPSM scoring model and FDR estimation paradigm developed in pGlyco2[10]. The scoring model used a linear mixture of peptide backbone and glycan structure evidence to score glycopeptides. The peptide score (Eq. (1)) was a mass accuracy weighted log-intensity summation, weighted by peptide sequence coverage (to exponent $\gamma$). The glycan score (Eq. (2)) followed the same pattern, save that the glycan coverage is broken into two terms, where the coverage along the entire topology is given one exponential weight $\alpha$, while the coverage of the conserved *N*-glycan core is given an additional exponent $\beta$. The two components are combined by a linear mixing weight $w$. Because the peptide and glycan scores were retained, the same mixture model-based FDR estimation procedure is applicable, allowing us to do a direct comparison with the results published in ref. 7. We used $\alpha = 0.5$, $\beta = 0.4$, $\gamma = 1$ and $w = 0.65$ for all variations of this scoring model, selectable as *log_intensity_reweighted* from the CLI.

$$\text{Score}_P(\gamma, \text{tol}) = \left[ \sum_i^{m_p} \log(\text{inten}_i) \times \left( 1 - \left| \frac{\text{ppm}_i}{\text{tol}} \right|^4 \right) \right] \times \text{coverage}_P^\gamma \quad (1)$$

$$\text{Score}_G(\alpha, \beta, \text{tol}) = \left[ \sum_i^{m_g} \log(\text{inten}_i) \times \left( 1 - \left| \frac{\text{ppm}_i}{\text{tol}} \right|^4 \right) \right] \times \text{coverage}_G^\alpha \\ \times \text{coverage}_{G,core}^\beta \quad (2)$$

We added a precursor mass accuracy bias (Eq. (3)) with $\mu_{pre} = 0$ and $\sigma_{pre} = 5$ ppm to prefer solutions with better precursor mass matches, given equal fragmentation evidence, exploited in ref. 14. We also included a penalty when a signature ion is expected for a monosaccharide but not observed or when a signature ion is observed without being expected (Eq. (6)) similar to ref. 47, in this work only common sialic acids were considered, but other abundant modified

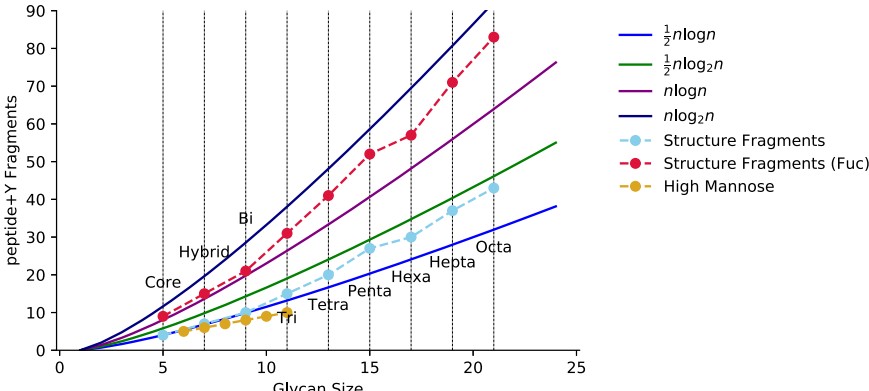

**Fig. 4 | Glycan composition fragment counting approximation.** The number of distinct mass fragments produced by fragmenting a known topology of *N*-glycan of a given size, enclosed by our two approximation proposals, with and without the presence of core fucosylation. Each labeled vertical line denotes a class of complex-type *N*-glycan of ascending number of lactosamine units beyond the core motif, which are arranged as separate branches. The immature high mannose from Man9 to Man4 are also shown, which fall below the approximation due to the homogeneity of their Y fragments.

monosaccharides are also available. This complete scoring function is expressed in Eq. (7).

$$\text{MassAcc}(\text{ppm}_{pre}, \mu_{pre}, \sigma_{pre}) = -10 \log_{10}\left(1 - \exp\left\{-\frac{(\text{ppm}_{pre} - \mu_{pre})^2}{2\sigma_{pre}}\right\}\right) \tag{3}$$

$$\text{UnexpIon}(o) = 10 \log_{10}\left(1 - \frac{\text{inten}_o}{\max \text{inten}}\right) \tag{4}$$

$$\text{MissIon}(o) = 10 \log_{10}\left(1 - \min\{o/2, 0.99\}\right) \tag{5}$$

$$\text{SignIon} = \sum_{o \in (g[\text{NeuAc}], g[\text{NeuGc}])} \begin{cases} \text{UnexpIon}(o) & o = 0 \\ \text{MissIon}(o) & o > 0 \text{ and } \frac{\text{inten}_o}{\max \text{inten}} \le 0.01 \\ 0 & \text{otherwise} \end{cases} \tag{6}$$

$$\text{Score}_{GP}(\alpha, \beta, \gamma, \text{tol}, w) = w \times \text{Score}_P(\gamma, \text{tol}) + (1 - w) \times \text{Score}_G(\alpha, \beta, \text{tol}) + \text{SignIon}() + \text{MassAcc}(\text{ppm}_{pre}, 0, 5 \times 10^{-6}) \tag{7}$$

**Glycan coverage approximation.** One advantage of pGlyco's method is that it is able to compute a formal coverage ratio for the glycan component by using the peptide+Y ion ladder and an exact enumeration of the theoretical fragments for each of their glycan topologies. This comes at the cost of requiring a topology for each glycan to be searched, expanding the search space to consider, despite lacking diagnostic fragmentation to discriminate between most topological isomers. We introduce a method for approximating the total number of theoretical fragments a glycan composition could generate if its monosaccharides were arranged in a tree structure. *N*-glycans are branching structures, similar to binary trees. The height of a balanced binary tree with *n* nodes is $\log_2 n$. Because peptide+Y fragment generation often involves cleavage events along multiple branches, we can assume an upper bound for fragments of a binary tree to be $n \log_2 n$. *N*-glycans are not truly binary trees: the unfucosylated core motif's root node has a single child node, suggesting the upper bound $\frac{n}{2} \log_2 n$ for *N*-glycans without core fucosylation or xylosylation. Beyond the first fan-out from the core motif, *N*-glycans are usually linear, causing the $n \log_2 n$ approximation to be too harsh, especially for large glycans. A

change to the natural log $n \log n$ turns out to be a close bound for small glycans and forgiving of large glycans which an exact coverage-based method is more stringent for. A comparison of the different rates of growth and divergence is shown in Fig. 4. This allows us to generate a coverage metric for glycan compositions, letting us use a more compact glycan composition database rather than a glycan structure database.

We generated semi-structured peptide+Y ion ladders from glycan compositions by explicitly generating fragments assuming that a core motif is present, marking these fragments as *core* fragments, possibly with deoxyhexose or pentose side-chain, and then adding every combination of remaining non-labile monosaccharides to the core motif, including biosynthetically improbable ones. We treated NeuAc and NeuGc as labile. To calculate glycan coverage, we first approximated the "size" of a glycan composition as the number of monosaccharide residues in the glycan composition minus the number of NeuAc and NeuGc, and deduct one if two or more dHex/Fuc residues are present, calling the final number $n_g$. We then used the approximation shown in Fig. 4, computing the normalizing factor $d_g$ (Eq. (9)).

$$n_g = |g| - g[\text{NeuAc}] - g[\text{NeuGc}] - (g[\text{dHex}] > 1) \tag{8}$$

$$d_g = \max\left(\begin{cases} n_g \log n_g & g[\text{dHex}] > 0 \\ \frac{1}{2} n_g \log n_g & g[\text{dHex}] = 0 \end{cases}, n_g\right) \tag{9}$$

Where $g[m]$ is the number of monosaccharide $m$ units in $g$ and $|g|$ is the cardinality of $g$, the number of discrete monosaccharide residues in the glycan composition. The coverage$_{G,core}$ is readily calculable as our algorithm explicitly enumerates the core fragments, while coverage$_G$ is the number of distinct peptide+Y fragments matched divided by $d_g$. Both target and decoy glycans were treated the same way, save that decoy glycans' peptide+Y fragments beyond Y1 were given a random mass shift between 1.0 and 30.0 Da drawn from a uniform random distribution. We treated *O*-glycans identically, noting that for $n_g \le 7$, $\frac{1}{2} n_g \log n_g < n_g$, so by Eq. (9) $d_g = n_g$. Many individual mucin *O*-glycans are less than 7 monosaccharides. No glycans considered in our *O*-glycome were above this threshold.

**MS2 FDR Estimation**

We estimated FDR as described in[14], except that the peptide FDR estimation procedure uses a semi-supervised linear SVM as described in[48,49] using `scikit-learn`[37]. For the base scoring model, the features used are the peptide score (Eq. (1)) and peptide coverage (*Coverage$_P$*),

and for the fragmentation prediction scoring model, the features used are the peptide score (Eq. (12)), the peptide coverage ($Coverage_P$), and the peptide fragment correlation $Cor'_P(\theta)$ in Eq. (11). We explicitly did not calculate the glycan FDR for glycan compositions of size 3 or smaller as these produce too few peptide+Y ions to be meaningful.

Following MSn FDR estimation, we localized all PTMs, including glycans, using an implementation of PTMProphet[50]. For each spectrum, all inferior localization solutions were removed from consideration for subsequent steps of aggregation and analysis.

## MS1 Scoring

After MSn spectra were assigned and FDR-controlled, we extracted all deconvoluted MS1 peaks from the processed MS data file and constructed MS1 features as described in ref. 26, save that the charge state component is set to a constant 0.8 and no adduct scoring was performed.

## Adduct deconvolution and retention time modeling

See supplementary section 11.1 for details on the adduct deconvolution process and subsequent retention time model building and online glycan composition revision process.

## Glycopeptide fragmentation modeling

**Inter-peak relationships.** Glycopeptide fragmentation is complex, including multiple charge states for the same theoretical fragment and the presence of both glycosylated and unglycosylated versions of peptide backbone fragments occurring interspersed. Many others[20,21,51] have demonstrated that a peak-fragment ion match, which is supported by related peak-fragment ion matches is worth more than a peak matched in isolation. We used a Bayesian probability model based on UniNovo's[21]. In addition to the link features described in the original method, we added a mass difference feature for HexNAc (203.0794 Da) for peptide backbone fragments as well as for HexNAc, Hexose (162.0528 Da) and dHex (146.0579) for peptide+Y ion series matches. We did not include neutral losses of NH3 or H2O and iterative refinement here, though they may be useful for future work. During training, we did not consider any peaks with an intensity rank below 1 but set no restriction on rank during inference.

UniNovo models multiple partitions over theoretical precursors independently by precursor mass as a proxy for peptide length assuming that fragmentation propensities for these structures will differ. As glycopeptides have both a peptide and a glycan component and a larger range of charge states than bare peptides, we use a multi-dimensional partitioning by peptide length, glycan size, precursor charge, precursor proton mobility[52], and the type and number of occupied glycosylation sites, with the ranges defined in supplementary Table 5. This produces up to 150 partitions per glycosylation type, though not all are expected to be populated.

We extracted GPSMs passing a 1% FDR threshold and a total score threshold of 20 from all samples in each dataset, converted to an annotated MGF format. For the mouse tissue dataset, we chose to reserve the brain tissue subset as a test set and fit our model on the remaining tissue types to demonstrate model performance and ability to generalize. For the human serum dataset, there was no obvious distinction between samples, so we used all samples for training to demonstrate the effect of whole dataset modeling for a large study. We partitioned GPSMs according to the rules described above, though in order to smooth over small numbers of observations in some groups, we mixed adjacent charge state groups while holding all other constraints constant, and fit our model for each ion series.

For each glycopeptide fragment match $f$ we compute the posterior probability of that peak using its series and the set of unique peak-pair features, which we term the reliability of the fragment match $\phi_f$.

**Peak intensity prediction.** A glycopeptide under collisional dissociation fragments in both the glycan and the peptide, with preference to weaker bonds breaking with greater frequency, subject to the physiochemical properties of the molecule and the activation energy used[53,54]. Prediction of whether a fragmentation event should generate an abundant peak has repeatedly been used as a method for improving peptide identification[23,25,51,52].

The mobile proton hypothesis is a widely accepted kinetic model of fragmentation for protonated peptides[53,55] which has been used to create many peptide fragmentation prediction algorithms[51,52]. Unsurprisingly, glycopeptide fragmentation depends on mobile protons as well, driving very different abundance patterns of fragmentation depending upon charge state[6,54]. We used the proton mobility classification scheme described in ref. 52, where the number of $K$, $R$, and $H$ are compared to the precursor ion's charge, where if the sum is greater than the charge, the precursor is *immobile*, if equal, the precursor is *partially mobile*, or less than, the precursor is *mobile*. We included this observation in the partitioning scheme we derived from UniNovo earlier and applied the same partitioning scheme when modeling relative intensities.

We modeled the relative intensity of a fragmenting glycopeptide as a probability drawn from a multinomial distribution parameterized by a set of features listed in supplementary Table 6. The features chosen were based upon the approaches described in refs. 51,52. It has been made clear that more sophisticated modeling techniques may be applied to this type of problem[23,25], but they lack interpretability and require substantial numbers of observations to train.

For each partition of the training data, without allowing sharing between adjacent charge states, we estimated the parameters of the multinomial distribution from glycopeptide spectrum matches using iteratively re-weighted least squares, weighted by the total signal in each spectrum, with the individual peaks weighted by the reliability $\phi$ using the peak relation model for that partition, or **1** if this lead to an unstable solution. For each partition, we fit a model on all of the GPSMS for predicting peptide backbone fragment intensities. Next, we split the GPSMS into two groups based on whether the most abundant peptide+Y ion series ascends or descends in intensity as the glycan composition grows in size, and fit a model on each for use when predicting peptide+Y fragment intensities for observed matches that show the same size-intensity trend. After this step, each partition contained a peak relation model $\phi$, a peptide relative intensity model, and two glycan relative intensity models $\boldsymbol{\theta}$.

**Integrating fragmentation modeling into scoring functions.** We incorporated peak relation-based reliability and peak intensity prediction into the glycopeptide scoring model's two moiety-specific branches. For each theoretical GPSM, we found the appropriate partition's models, or the nearest partition if it were missing.

The prediction-enhanced scoring model extended the base scoring model with new components. For the peptide score shown in Eq. (12) the fragment-level reliability $\phi$ channels weight away from lower confidence peaks, diminishing their influence on the score, while the correlation term $Cor_P(\theta)$ gives a benefit to matches that match more fragments while still correlating with the intensity model (Eq. (10)). For Eq. (10), we added a shifted Pearson correlation $\rho$ of the observed and predicted intensities of peptide fragments scaled by $\log_{10} m_p$ where $m_p$ is the number of peptide fragments matched to prefer solutions that match more peaks at the detriment of worse correlation, given the weak average correlation the peptide model has. We also found that the total reliability of all peaks matched was nearly as useful as the observed intensity itself. During FDR estimation, we explicitly include Eq. (11), which scales linearly with peptide backbone fragments normalized by coverage instead of logarithmically which we found provided better separation between targets and decoys. If the peptide

correlation were improved, the correlation shift wouldn't be needed, but the $m_p$ scaling factors might still be necessary because of the partial fragmentation of peptide backbones under SCE. Under different dissociation conditions that preserve more intermediate glycan fragmentation attached to the peptide backbone fragments, such as ETD or EThcD, it would require revisiting how reliabilities interact between peptide backbone fragments with partially fragmented glycans attached.

$$Cor_P(\theta) = \left(\rho\left(\text{inten}_P, \theta(\text{gpsm}_P)\right) + 1\right) \times \log_{10} m_p \quad (10)$$

$$Cor'_P(\theta) = \left(\rho\left(\text{inten}_P, \theta(\text{gpsm}_P)\right) + 1\right) \times m_p \times \text{coverage}_P \quad (11)$$

$$\text{ScoreModel}_P(\gamma, \text{tol}, \theta) = \left(\left[\sum_i^{m_p} \log_{10}(\text{inten}_i) \times \left(1 - \left|\frac{\text{ppm}_i}{\text{tol}}\right|^4\right)\right.\right.$$
$$\left.\left. \times (\phi_i + 1) + \phi_i\right] + Cor_P(\theta)\right) \times \text{coverage}_P^\gamma \quad (12)$$

The model glycan score, shown in Eq. (17) is more complex. For peptide+Y ions, the reliability $\boldsymbol{\phi}$ and $Cor_G(\theta)$ are consistently larger but could be skewed for observations with a small number of matched fragments but strong agreement with the model. To counter this, we scaled these quantities by the number of non-trivial glycan fragments that have been observed, Eq. (13), sigfrag$_g$. We observed scenarios where the model chose different peptide backbones to favor low proton mobility when incomplete fragmentation was available, often abandoning abundant peptide backbone fragments to do so. Given that the glycan-describing fragments depend upon the peptide sequence being correct via the peptide backbone composition features, but do not themselves help define the peptide backbone beyond the crude mass aggregate, we limited the model's contribution to the glycan score by scaling it by Eq. (14), a function of peptide coverage $s(\text{coverage}_P)$, where $s$ was chosen from a family of exponential functions clamped to the [0, 1] range within the domain of [0, 1]. Similarly, we limited the contribution of the reliability by glycan correlation to prevent that term from selecting worse solutions that happen to match an extra random peak as shown in Eq. (15) where the total reliability is scaled down by a function of the glycan intensity correlation. Because reliabilities are often quite high for glycan fragments, we did not scale intensity by reliability, letting us factor the term out of the first sum and scale it prior to combining it with the base score. If glycan fragmentation were more sparse, it might be favorable to change this to follow the pattern used in Eq. (12).

$$\text{sigfrag}_g = \sum_i^{m_g} \text{fragment glycan size}_i > 1 \quad (13)$$

$$\text{pad}(x) = 0.5 \times x + 0.5$$

$$s(x) = \min(\exp(3 \times x - 1), 1) \quad (14)$$

$$Rel_G(\theta) = \sum_i^{m_g} \text{pad}(\phi_i) \times \max\left(\rho(\text{inten}_G, \theta(\text{gpsm}_G)), 0.25\right) \quad (15)$$

$$Cor_G(\theta) = \frac{\rho(\text{inten}_G, \theta(\text{gpsm}_G)) + 1}{2} \quad (16)$$

$$\text{ScoreModel}_G(\alpha, \beta, \text{tol}, \theta) = \left(\left[\sum_i^{m_g} \log_{10}(\text{inten}_i) \times \left(1 - \left|\frac{\text{ppm}_i}{\text{tol}}\right|^4\right)\right] + \right.$$
$$\left[Rel_G(\theta) + Cor_G(\theta)\right] \times \text{sigfrag}_g \times s(\text{coverage}_P)\right) \times$$
$$1em\text{coverage}_G^\alpha \times \text{coverage}_{G,core}^\beta \quad (17)$$

We combine Eqs. (12) and (17) in a linear mixture in Eq. (18), along with two auxiliary scores to help select a more parsimonious solution.

$$\text{ScoreModel}_{GP}(\alpha, \beta, \gamma, \text{tol}, w, \theta) = w \times \text{ScoreModel}_P(\gamma, \text{tol}, \theta) + (1 - w) \times$$
$$\text{ScoreModel}_G(\alpha, \beta, \text{tol}, \theta) + \text{SignIon}() + \quad (18)$$
$$\text{MassAcc}(\text{ppm}_{pre}, 0, 5 \times 10^{-6})$$

**Site-specific glycome network smoothing**

We and others have shown that it is useful to exploit the relationships between biosynthetically nearby glycans when evaluating glycan confidence[26,56]. Briefly, the method described in ref. 26 constructs a graph over the glycome, where nodes are glycan compositions and edges are addition/removal of a monosaccharide, and divides it into biosynthetic neighborhoods. Next, it maps observed glycan compositions onto that graph and estimates coefficients $\boldsymbol{\tau}$ of those biosynthetic neighborhoods and the glycans within them, both observed and unobserved, including a mechanism by which information from observed glycans $\boldsymbol{\phi}$, which we will refer to as **u** here as $\phi$ is already used, may be propagated to unobserved ones via a smoothing parameter $\lambda$. In effect, $u_g$ is a function of the observed score for previously seen glycan composition $g$, the weighted degree of $g$ in the glycome graph, and a weighted inner product over the subset of $A_g^t \cdot \boldsymbol{\tau}$ that $g$ overlaps, scaled by $\lambda$.

We extended the method for released glycans we presented in ref. 26 to the glycoforms observed at each glycosite of a glycoprotein, using high-confidence identified glycopeptides spanning those sites, treating each site independently. For all datasets, only glycopeptides with MS1 scores > 0 and passing a joint FDR threshold of 1% were considered. For the mouse tissue dataset, we fit site-specific glycome models for each tissue type separately. For the human serum datasets, we used the full dataset to estimate site-specific glycome models. The human serum dataset used the neighborhood rules found in ref. 26, while the mouse tissue datasets used an extended set of neighborhood bounds to take into account biosynthetic pathways absent in humans and other old-world primates[1]. For more details on the site-level parameter estimation process, see supplementary Section 11.5.

**Integrating network smoothing into scoring functions.** To incorporate the **u** into the search process, when we generated theoretical glycopeptides, we determined their protein and spanned glycosylation site(s) of origin and looked up the appropriate $u_g$ for that glycopeptide's glycan $g$ from that site's model. Decoy glycans were treated identically to their target counterparts, given the same value for $u_g$. Decoy peptides are looked up against decoy proteins whose site models are the same as their target counterparts' reflected along the reversed sequence to align with the same residue and also behave identically.

We incorporated $u_g$ into the scoring function by adding it to the total evidence before being scaled by glycan coverage, as shown in Eqs. (20) and (19), where $u_g = 0$ if no site model was fit for that glycosite. We applied this technique to only $N$-glycosylation sites no models were fit for $O$-glycosylation sites.

$$\text{ScoreSmoothed}_G(\alpha, \beta, \text{tol}) = \left(\left[\sum_i^{m_g} \log_{10}(\text{inten}_i) \times \left(1 - \left|\frac{\text{ppm}_i}{\text{tol}}\right|^4\right)\right] + u_g\right)$$
$$\times \text{coverage}_G^\alpha \times \text{coverage}_{G,core}^\beta \quad (19)$$

**Table 1 | Software versions**

| Library | Version |
|---|---|
| glycresoft | 0.4.23 |
| glycopeptide_feature_learning | 0.1.0 |
| ms_deisotope | 0.0.53 |
| ms_peak_picker | 0.1.46 |
| glypy | 1.0.13 |
| glycopeptidepy | 0.0.27 |
| psims | 1.3.4 |
| pyteomics | 4.6.3 |
| lxml | 4.6.4 |
| numpy | 1.22.4 |
| scipy | 1.7.2 |
| scikit-learn | 1.1.2 |

The versions of Python libraries used.

$$\text{ScoreModelSmoothed}_G(\alpha, \beta, \text{tol}, \theta) = \left( \left[ \sum_i^{m_g} \log_{10}(\text{inten}_i) \times \left( 1 - \left| \frac{\text{ppm}_i}{\text{tol}} \right|^4 \right) \right] + \right.$$
$$\left[ Rel_G(\theta) + Cor_G(\theta) \right] \times Signif_G \times s(\text{coverage}_P)$$
$$\left. + u_g \right) \times \text{coverage}_G^{\alpha} \times \text{coverage}_{G,core}^{\beta}$$

$$(20)$$

## Libraries used

GlycReSoft is written in Python and Cython[57] and uses NumPy[58], SciPy[59], MatPlotLib[60], and scikit-learn[37]. Library versions used in this publication are listed in Table 1.

## Reporting summary

Further information on research design is available in the Nature Portfolio Reporting Summary linked to this article.

## Data availability

The original datasets re-analyzed in this study are available from the PRIDE archive under the following accession numbers: mouse PXD005411 (brain), PXD005412 (kidney), PXD005413 (heart), PXD005553 (liver), and PXD005555 (lung), and fission yeast PXD005565 for[10], PXD005931 for[31], and PXD009654 for[32]. The results from this work are provided in the supplementary materials available on FigShare at https://doi.org/10.6084/m9.figshare.24578857. Source data are provided with this paper.

## Code availability

The source code for the search engine and network smoothing algorithm are part of GlycReSoft is available at https://github.com/mobiusklein/glycresoft. The fragmentation modeling codebase is available at https://github.com/mobiusklein/glycopeptide_feature_learning. Both codebases are released under the Apache 2 License.

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

## Acknowledgements

Financial support was provided by NIH grant R35GM1344090 (J.Z.)

## Author contributions

Conception and design: J.K. and J.Z. Developed methods: J.K. and L.C. Collected data and performed the analysis and interpretation: J.K. Writing, review, and/or revision of the manuscript: J.K., J.Z., and L.C. Study supervision: J.Z.

## Competing interests

The authors declare no competing interests
