## [Transparent Peer Review file · Nature Communications]

Expanding N-Glycopeptide Identifications By Modeling Fragmentation, Elution, and Glycome Connectivity

Corresponding Author: Dr Joshua Klein

Version 0:

Reviewer comments:

Reviewer #1

(Remarks to the Author)

I have paid attention to this project on GitHub for a long time, it is a nice open-source and python-based project. This work combined two interesting methods to improve glycopeptide identification: glycopeptide spectrum prediction and glycome network smoothing. And they seem to work quite well.

Glycopeptide spectrum prediction will be of course useful in glycopeptide search engines, as researchers have been using spectrum prediction in the peptide search engine, peptide de novo sequencing, etc. It will be interesting to use it in the glycopeptide search engine. However, the prediction method is not good enough (median $R \leq 0.8$ in mouse datasets). It may be because the authors used the mobile proton hypothesis which was applied in UniNovo. But authors should compare this model with machine learning-based methods such as XGBoost. The mobile proton model is not easy to be formalized, making it difficult for the practical use. On the contrary, machine learning models does not depend on the physical hypothesis, hence I think it may have good performance. So I think the comparison is quite necessary. Machine learning (random forest, XGBoost, feed-forward neural network, or even deep learning) has shown great improvement in spectrum prediction, and it is not difficult to use in the Python environment. Otherwise, it could convince me if the authors could prove that more accurate spectrum prediction is not necessary for glycopeptides.

Glycome network is also attractive. And I believe it could be not only used just as a factor in the scoring function. However, this part is difficult to read. The details about how to construct the network and calculate the smoothing are all described in ref[24], but I think authors should add some key description texts in this manuscript for readers. For example, τ is very important to understand Fig. 7, but the authors did not explained what did τ mean, making it difficult to read Fig. 7. Although τ is described in ref[24], but it is not easy to remember the math formula after switching from ref[24] back to this paper for readers.

In glycome network smoothing, it seems that decoy-glycans are treated differently from target-glycans, it will result in different score distributions for decoy-glycans and false target-glycans. The authors need to test it if it does affect the FDR estimation.

There are too many formulas for scoring functions in the manuscript, most of them are not necessary to be listed in the main manuscript, I suggest to keep some key formulas and move others into Supplementary File.

The workflow of GlycReSoft is not clear, I believe using a figure to describe the workflow will be better for understanding, especially for readers with wet-lab background.

Supplementary scores were described but not compared with the base score.

$u_{\{gpsm\}}$ in eq.22 is not described in the main texts.

Details of Table 2 should be described in the Supplementary File, as it is very complicate.

In Fig.8, retained GPSMs of "peptide FDR < 1%" should not be less than "joint FDR < 1%", because "joint FDR < 1%" implies "peptide FDR < 1%" and "glycan FDR < 1%".

Private communication should not be a reference as readers could not read it (ref[39]).

It would be better to show one or two examples about how spectrum prediction and glycome network improve the identifications. For example, the glycoPSM was >1% FDR, but its "neighbor glycans" improve its score (<1% FDR) using glycome network.

The authors have developed RT prediction tools for glycopeptides, why do they not use it to improve the glycopeptide score in this paper?

I think glycome network would be very useful to improve the glycopeptide identification using MS1 signals. For a given MS2-identified site-specific glycan, one can search its "neighbor glycans" in MS1, and hence obtains more site-specific glycans, especially for purified glycoproteins. It may be also useful for quantification.

Python 2.7 is not officialy supported anymore, is there any plan to upgrade the Python version for users? Sometimes users would like to integrate GlycReSoft in their own python workflows.

Dr. Wen-Feng Zeng

Reviewer #2

(Remarks to the Author)

In this manuscript, Klein et al. describe improvements to their previously published GlycReSoft software package for annotating intact glycopeptide tandem mass spectra. Glycoproteomics is a rapidly growing and important field that has a critical need for improved data and statistical analysis tools, making the submitted manuscript of significant interest. This manuscript describes the addition of glycopeptide spectrum prediction and an update to the glycan network modeling previously described, as well as the reanalysis of existing data to compare the performance of GlycReSoft with these new methods to other glycoproteomics software packages.

The new methods are well described technically and clearly improve the annotation of glycopeptide spectra, enabling GlycReSoft to annotate more spectra than its previous iteration and pGlyco2.0 (another software package with similar search and scoring methods). The spectrum prediction method appears to be a promising foray into a challenging area, though the glycan network smoothing method would benefit from additional discussion to clearly distinguish it from the previous iteration. In addition, there are several areas of the manuscript that need significant revision prior to publication, most notably the comparison with MSFragger. The manuscript is heavily focused on the technical details of the algorithms, which are valuable and of great interest to readers in (glyco)proteome informatics, but it needs additional attention to the discussion of results and quality of figures to be of interest to a broad audience.

Key Revisions:

- The results section is extremely short and includes some subsections that simply restate information presented in the associated figure(s). Importantly, several key results are not discussed in sufficient detail, despite additional attention to some of the results in the subsequent discussion section. For example:
 - o Section 3.3/4.1: The far greater reliability of Y (peptide+glycan) ions than b or y (peptide) ions is noted as confirming previous work, but the implications for prediction accuracy are not discussed in sufficient detail. In particular, because Y ions do not contribute to peptide identification, could the strong preference to model them accurately instead of peptide b/y ions decrease the accuracy of peptide identification? Also, it would be very useful to see these trends as a function of charge state in addition to aggregate to support the discussion.
 - o Section 3.5/4.3: Why are smoothing parameter distributions bimodal for many glycan classes? The text describes the importance of glycans with $T > 1$ in several cases where there is also a large population of the class with $T \sim 0$. Are there any features of the glycans that have $T \sim 0$ in the high mannose class (for example) that distinguish them from those in the class that have $T > 10$? What does $T > 1$ mean in an intuitive sense?
 - o Section 3.2: What is the overlap in gPSMs found in the original analysis vs with fragmentation modeling and glycan network smoothing? What is the overlap with the pGlyco2 analysis?
- The glycan network smoothing presented builds upon a previously published method. Additional discussion should be added to clearly distinguish the method presented here from the older method. A description of the old method and any related work should be added to the introduction. The data presented in Figures 2-3 only compares the performance of the

network smoothing model in combination with fragmentation modeling. A comparison to the older method using the new network smoothing alone (not using the fragmentation model) should be added to aid in distinguishing the gains attributable to the smoothing model specifically.

- The glycan network smoothing uses external knowledge to improve annotation of spectra, as the authors note. This likely limits its application to systems with well-defined glycosylation profiles. In addition, even in systems with “well-defined” glycosylation now, it is well known that enrichment methods for glycopeptides used to generate site-specific glycosylation information are biased towards specific types of glycans. The authors should note this and comment on whether this can impact spectral annotation of glycans from non-model organisms or enriched with less common methods, and if there are cases in which it should not be used as a result.

- Section 4.4 discusses another reanalysis of the same data with another recently published software tool, MSFragger. The authors note that MSFragger uses a different scoring function and FDR method and obtains more spectral annotations than the analysis presented in this manuscript. In my view, it is not necessary for the authors to obtain as many gPSMs as the MSFragger analysis for this manuscript to be worthy of publication. The methods for filtering potential glyco-spectra, scoring the fragmentation of glycopeptides, and, as the authors note, computing FDR are different, and thus obtaining different results is not unexpected. However, the manuscript presents a misleading analysis of FDR rates and has several areas that need to be revised:

- o The authors appear to claim that the difference in number of PSMs annotated by GlycReSoft vs MSFragger is entirely the result of different FDR methods. This should be revised to note the many differences in ion types considered in scoring, methods for determining which spectra to search for glycopeptides, and different outcomes controlled at 1% (annotation of a peptide with a specific glycan vs annotation of a peptide and a mass shift) between the tools, all of which likely contribute substantially to the differences in number of PSMs obtained.

- o The analysis presented in Figure 8 requires revision. The union of multiple analyses of the same data with individual 1% FDR rates does not itself have a 1% FDR. True positives are more likely than false positives to generate high scores and thus are more likely to pass a given confidence threshold, making a given true positive more likely than a false positive to be accepted in each individual analysis of the data (and thus to be in common between them). When the union of these individual analyses is taken, the true positives are thus more likely to be redundant than false positives, leading to an effective enrichment of false positives in the union, increasing the FDR above 1% (possibly very far above 1%, depending on the analyses in question). As such, the “any FDR <1%” category, a union of 3 result sets, does not maintain the 1% FDR of its component sets. It should not be compared to other analyses that use a 1% FDR and should be removed from Fig. 8.

- o The authors state that they are able to obtain a similar number of PSMs as the MSFragger analysis when considering their union of PSMs passing any of FDRs, or over 60,000 PSMs when considering the union and applying glycan network smoothing. However, as noted above, the union is no longer at 1% FDR and is not comparable to an analysis that is at 1% FDR. These statements should be removed.

- o Additional discussion of the peptide-only FDR result is needed. The authors contention that the additional PSMs obtained by MSFragger can be attributable to the different FDR is undercut by the fact that the peptide-only FDR is the most similar to MSFragger’s method, but is by far the most conservative metric in Figure 8, obtaining far fewer PSMs than the joint FDR method. How was this peptide-only FDR calculated?

- o The authors state that the “bin-specific probabilities in PeptideProphet may be learned from a small number of observations” for a given mass shift, which is not correct. The mass model used in PeptideProphet models the probability as a result of the number and score of PSMs in each mass bin relative to the number and score of PSMs in other bins, as well as the proportion of targets and decoys in that bin. In a mass bin with few PSMs, the resultant probability is primarily a result of the low number of PSMs relative to other bins, meaning the mass bin probabilities are modeling using information from the complete dataset even if there are only a few PSMs in a given bin. This paragraph should be revised to reflect this.

- Some figures need graphical revisions:

- o Table 2: The features used to predict spectra are not clearly defined, and the mass of them presented in the table (while somewhat visually impressive) does not add much to understanding.

- Terms used should be defined either in the table or supporting information (e.g. what is “stub glycopeptide”?).

- The table could also be greatly simplified by combining some iterative entries (e.g. glycan loss 0-20 instead of listing glycan loss 0, glycan loss 1, ...).

- o Figure 6: The goal of the figure is to show similarity between observed and predicted spectra, but the visual display of the data makes this very hard to assess.

- All spectra need to be increased in size, especially in the vertical dimension to aid in distinguishing the heights of peaks (perhaps tile all 4 vertically rather than a 2x2 grid)

- Overlapping text labels for peaks need to be moved so they can be read

- Text labels need to be moved far enough away from peaks that peak height can be accurately distinguished

- All text needs to be increased in size to be visible without requiring excessive zooming. If this is not feasible for all individual peaks, perhaps a visual representation of the ion types could be substituted for long text labels.

Reviewer #3

(Remarks to the Author)

The manuscript by Klein et al., is an impressive update to the GlycReSoft glycoproteomic bioinformatics pipeline from the Zaia research group. The manuscript describes an informatic approach which increases the assignment of glycopeptides in complex MS datasets by cross-referencing the measured values of this dataset with an in-silico search space that is guided by a hybridisation/combination of several algorithms such as a glycopeptide spectral matching algorithm, a glycan coverage

approximation, inter-peak relationships, peak intensity predictions and a glycome network smoothing. Whilst much of these approaches have been reported in previous publications, several improvements to approaches are discussed and presented in comprehensive set of scoring algorithms. The combination of approaches are well thought out in-silico approach that increases the number of confident assignments whilst keeping the FDR low. The statistics do support the improvements.

Whilst the paper is impressive, it is very heavy on the algorithms and mathematical approach and subsequent validation with previously generated datasets. As such, the findings here may be lost on the audience who would most benefit from this - glyco-analytical scientists. The reviewer has therefore some queries and guiding comments with the intention to helping the paper to be more convincing to an audience who would most benefit from this manuscript and yet may be overwhelmed by the informatic approach;

- is the calculated FDR the only measure of accuracy we have for this dataset? how are we certain the improvements to the assignments are real outside of your informatic approach? would it be feasible to demonstrate your approach with additional experiments with exoglycosidases to demonstrate the efficacy of your assignments?

- can you comment on what extra glycopeptides you were able to identify from the comparison of the base model with the glycan network smoothing fragmentation model on these datasets?

- the dataset which you used was generated using an orbitrap fusion with stepped collision in HCD. whilst this is an ideal situation, there does remain the question of how well your approach would be to other types of mass spectrometry formats. For example, can you demonstrate any gains or losses of your approach to time of flight mass spectrometers? Would the beam type formats of mass spectrometers would affect the peak intensity prediction?

- can you explain why you didn't consider the oxonium ions into your scoring method when in fact, these ions are very valuable diagnostic in glycopeptide identification?

- some of the figures such as figures 4, 5, 6 are incredibly hard to read and are of very low resolution. Since they are an integral part of your thesis, could you please provide higher resolution images for this manuscript

Author Rebuttal letter:

We have rewritten much of the content of the manuscript to address the reviewers's concerns, including development of new methods.

REVIEWER COMMENTS

Reviewer #1 (Remarks to the Author):

I have paid attention to this project on GitHub for a long time, it is a nice open-source and python-based project. This work combined two interesting methods to improve glycopeptide identification: glycopeptide spectrum prediction and glycome network smoothing. And they seem to work quite well.

Glycopeptide spectrum prediction will be of course useful in glycopeptide search engines, as researchers have been using spectrum prediction in the peptide search engine, peptide de novo sequencing, etc. It will be interesting to use it in the glycopeptide search engine. However, the prediction method is not good enough (median $R \leq 0.8$ in mouse datasets). It may be because the authors used the mobile proton hypothesis which was applied in UniNovo. But authors should compare this model with machine learning-based methods such as XGBoost. The mobile proton model is not easy to be formalized, making it difficult for the practical use. On the contrary, machine learning models does not depend on the physical hypothesis, hence I think it may have good performance. So I think the comparison is quite necessary. Machine learning (random forest, XGBoost, feed-forward neural network, or even deep learning) has shown great improvement in spectrum prediction, and it is not difficult to use in the Python environment. Otherwise, it could convince me if the authors could prove that more accurate spectrum prediction is not necessary for glycopeptides.

Thank you for the suggestion. We acknowledge that the mobile proton model of peptide fragmentation is difficult to apply in a useful manner, especially as longer range effects and higher charge multiplicities are considered. We were concerned that a more complex model would be too vulnerable to over-fitting given the limited data available. By refining our model we were able to improve glycan correlation above 0.818 on the test dataset, shown in Figure 2g of the main text. We addressed this concern by training a gradient boosting forest regression using a method similar to MS2PIP, using scikit-learn in supplementary section S3, training on model for b ions and another for y ions for charge states +1 and +2. After combining the result of each model, we evaluated it on the train and test datasets of the mouse tissue dataset, finding that while ensemble did

substantially better on the training dataset with a PCC of 0.814 (compared to 0.536 with our model), but only performed slightly better on the test dataset with a PCC of 0.548 (compared to 0.501 with our model). Given this huge disparity between datasets, it seemed inappropriate to claim such superior performance and difficult to defend using a model that overfit so drastically.

We modified our peptide FDR estimation procedure to use a semi-supervised support vector machine which uses the aggregate peptide score, peptide backbone coverage, and a function of the peptide correlation and peptide backbone coverage as input features. We show in Figure 2h that for the majority of samples in the mouse tissue datasets, the peptide correlation-based feature was more valuable than the peptide backbone coverage feature alone, even in the brain samples which were not included during training.

Glycome network is also attractive. And I believe it could be not only used just as a factor in the scoring function. However, this part is difficult to read. The details about how to construct the network and calculate the smoothing are all described in ref[24], but I think authors should add some key description texts in this manuscript for readers. For example, \bar{I} is very important to understand Fig. 7, but the authors did not explained what did \bar{I} mean, making it difficult to read Fig. 7. Although \bar{I} is described in ref[24], but it is not easy to remember the math formula after switching from ref[24] back to this paper for readers.

Thank you for the suggestion. We have added a high level description of the method to provide an intuition for the reader in section 2.2.4, with a longer description in methods section 5.7 and the full derivation in the supplementary material.

In glycome network smoothing, it seems that decoy-glycans are treated differently from target-glycans, it will result in different score distributions for decoy-glycans and false target-glycans. The authors need to test it if it does affect the FDR estimation.

We shared your concerns, and now state explicitly in the text that targets and decoys share the same site parameters, even preserving them after protein reversal, and that decoy glycans receive the same value of u_g as target glycans at the same site. To go further, we replicated the entrapment study from the pGlyco3 paper, showing that the entrapment experiment using a mouse tissue site-specific network smoothing model that gave no benefit to any yeast glycosites had essentially the same entrapment FDR as the GlycReSoft search without network smoothing, differing by a single GPSM out of >6,000, shown in Fig. 3b.

There are too many formulas for scoring functions in the manuscript, most of them are not necessary to be listed in the main manuscript, I suggest to keep some key formulas and move others into Supplementary File.

Thank you for this suggestion. We have moved more of the numerical definitions to the supplementary material and have shifted the flow of the text to introduce those formulae we did not move to the supplement to the end of the text.

The workflow of GlycReSoft is not clear, I believe using a figure to describe the workflow will be better for understanding, especially for readers with wet-lab background.

Thank you for this excellent idea. We've created a workflow diagram that describes each stage of GlycReSoft's use in Fig. 1.

Supplementary scores were described but not compared with the base score.

We included the supplementary scores for parity with our prior work, where they served to limit pathological scoring function behavior on other datasets.

$u_{\{gpsm\}}$ in eq.22 is not described in the main texts.

Thank you for catching this. We have rewritten this section and used the correct symbol, u_g

Details of Table 2 should be described in the Supplementary File, as it is very complicate.

Thank you for the suggestion, we have moved this table to the supplementary material.

In Fig.8, retained GPSMs of $\hat{\alpha}$ peptide FDR < 1% should not be less than $\hat{\alpha}$ joint FDR < 1%, because $\hat{\alpha}$ joint FDR < 1% implies $\hat{\alpha}$ peptide FDR <

1% and glycan FDR < 1%.

Thank you for catching this, there was indeed an error in the code that produced this figure. It has been removed as the narrative focus has shifted.

Private communication should not be a reference as readers could not read it (ref[39]).

We have removed the reference to this discussion.

It would be better to show one or two examples about how spectrum prediction and glycome network improve the identifications. For example, the glycoPSM was >1% FDR, but its neighbor glycans improve its score (<1% FDR) using glycome network.

Thank you for the recommendation. We provide an example in Figure 2j showing an MS2 spectrum that was rescued by network smoothing, and further expand on it in the supplementary Section 8 and Figure S12 which depicts a set of rescued glycoforms for the same glycosylation site on the MS1 level. The rescued glycoforms are consistent with the retention time model learned for the sample in the base search, which provides a form of internal validation, though it is not completely orthogonal.

The authors have developed RT prediction tools for glycopeptides, why do they not use it to improve the glycopeptide score in this paper?

Thank you for this excellent suggestion. We had originally felt that this work was sufficient on its own, and that our prior work with relative retention time prediction was tied to manually selecting gradient transition points that it would not be feasible to apply here. Following your recommendation, we completely redesigned our relative retention time model fitting procedure to automatically adapt to changes in gradient conditions and mis-assigned glycan compositions, discussed in sections 2.2.2 and 3.1. The method itself is quite involved, and the details are covered in supplementary section 1.1. Figures 1b, 1c, 1d, and 1e cover outcomes from this work for the mouse tissue dataset.

I think glycome network would be very useful to improve the glycopeptide identification using MS1 signals. For a given MS2-identified site-specific glycan, one can search its neighbor glycans in MS1, and hence obtains more site-specific glycans, especially for purified glycoproteins. It may be also useful for quantification.

We agree that expanding the search for neighboring glycans in MS1 would be very useful, but we do not think that such matches are anywhere close to as confident identifications as those with MS2 support. This is especially true for large and complex samples with complex glycosylation patterns. There are often several glycan compositions that could equally well explain an MS1 feature at any given time, even with the same peptide backbone. Without MS2, we cannot resolve that to a usefully small set of alternatives outside of some trivial cases. The network smoothing method serves to take those cases with MS2 spectra and bring them closer to being identifiable, and the retention time model can refine them further.

Python 2.7 is not officially supported anymore, is there any plan to upgrade the Python version for users? Sometimes users would like to integrate GlycReSoft in their own python workflows.

This is a good point. We have updated the program and all of its dependencies to be fully compatible with Python 3.8 through 3.10. We have dropped support for Python 2.

Dr. Wen-Feng Zeng Thank you again for all of your excellent feedback.
Reviewer #2 (Remarks to the Author):

In this manuscript, Klein et al. describe improvements to their previously published GlycReSoft software package for annotating intact glycopeptide tandem mass spectra. Glycoproteomics is a rapidly growing and important field that has a critical need for improved data and statistical analysis tools, making the submitted manuscript of significant interest. This manuscript describes the addition of glycopeptide spectrum prediction and an update to the glycan network modeling previously described, as well as the reanalysis of existing data to compare the performance of GlycReSoft with these new methods to other glycoproteomics software packages.

The new methods are well described technically and clearly improve the annotation of glycopeptide spectra, enabling GlycReSoft to annotate more spectra than its previous iteration and pGlyco2.0 (another software package

with similar search and scoring methods). The spectrum prediction method appears to be a promising foray into a challenging area, though the glycan network smoothing method would benefit from additional discussion to clearly distinguish it from the previous iteration. In addition, there are several areas of the manuscript that need significant revision prior to publication, most notably the comparison with MSFragger. The manuscript is heavily focused on the technical details of the algorithms, which are valuable and of great interest to readers in (glyco)proteome informatics, but it needs additional attention to the discussion of results and quality of figures to be of interest to a broad audience.

Key Revisions: ¶ The results section is extremely short and includes some subsections that simply restate information presented in the associated figure(s). Importantly, several key results are not discussed in sufficient detail, despite additional attention to some of the results in the subsequent discussion section. For example:

Thank you for the suggestion. The article has been re-written using a different structure, making the results section is more descriptive, also moving much of the unnecessary detail out of the main text.

o Section 3.3/4.1: The far greater reliability of Y (peptide+glycan) ions than b or y (peptide) ions is noted as confirming previous work, but the implications for prediction accuracy are not discussed in sufficient detail. In particular, because Y ions do not contribute to peptide identification, could the strong preference to model them accurately instead of peptide b/y ions decrease the accuracy of peptide identification? Also, it would be very useful to see these trends as a function of charge state in addition to aggregate to support the discussion.

This is a good question. We observed this as well, and in the rewrite, we introduced a new term in the fragmentation model glycan score in Eq. 18 in Section 5.6 on page 21 that scales the contribution of the glycan model features to the score by an exponential function of the peptide backbone coverage, defined in Eq. 15 of the same section, and it is discussed in the text. The figure the the latter part of this comment refers to has been removed from the main text entirely.

o Section 3.5/4.3: Why are smoothing parameter distributions bimodal for many glycan classes? The text describes the importance of glycans with $T > 1$ in several cases where there is also a large population of the class with $T \sim 0$. Are there any features of the glycans that have $T \sim 0$ in the high mannose class (for example) that distinguish them from those in the class that have $T > 10$? What does $T > 1$ mean in an intuitive sense?

Thank you for highlighting this. The distribution of T is now discussed in Section 2.2.4, and more expressively in Section 3.3.

o Section 3.2: What is the overlap in gPSMs found in the original analysis vs with fragmentation modeling and glycan network smoothing? What is the overlap with the pGlyco2 analysis?

Thank you for pointing out this deficiency. This is now shown in the figures in Section S7, Figures S9, S10, and S11 in the supplementary material with UpSet plots showing distinct and overlapping glycans, GPSMs, and glycopeptides across all search strategies used for the mouse tissue datasets. We would have liked to include these figures in the main article, but they use up too much space on their own.

¶ The glycan network smoothing presented builds upon a previously published method. Additional discussion should be added to clearly distinguish the method presented here from the older method. A description of the old method and any related work should be added to the introduction. The data presented in Figures 2-3 only compares the performance of the network smoothing model in combination with fragmentation modeling. A comparison to the older method using the new network smoothing alone (not using the fragmentation model) should be added to aid in distinguishing the gains attributable to the smoothing model specifically.

This is also covered in Section S7, Figures S9, S10, and S11.

¶ The glycan network smoothing uses external knowledge to improve annotation of spectra, as the authors note. This likely limits its application

to systems with well-defined glycosylation profiles. In addition, even in systems with a well-defined glycosylation now, it is well known that enrichment methods for glycopeptides used to generate site-specific glycosylation information are biased towards specific types of glycans. The authors should note this and comment on whether this can impact spectral annotation of glycans from non-model organisms or enriched with less common methods, and if there are cases in which it should not be used as a result.

This is a good point. We now mention the importance of picking an appropriate glycome and the consequences of an incorrect choice in Section 3.3.

Section 4.4 discusses another reanalysis of the same data with another recently published software tool, MSFragger. The authors note that MSFragger uses a different scoring function and FDR method and obtains more spectral annotations than the analysis presented in this manuscript. In my view, it is not necessary for the authors to obtain as many gPSMs as the MSFragger analysis for this manuscript to be worthy of publication. The methods for filtering potential glyco-spectra, scoring the fragmentation of glycopeptides, and, as the authors note, computing FDR are different, and thus obtaining different results is not unexpected. However, the manuscript presents a misleading analysis of FDR rates and has several areas that need to be revised:

- o The authors appear to claim that the difference in number of PSMs annotated by GlycReSoft vs MSFragger is entirely the result of different FDR methods. This should be revised to note the many differences in ion types considered in scoring, methods for determining which spectra to search for glycopeptides, and different outcomes controlled at 1% (annotation of a peptide with a specific glycan vs annotation of a peptide and a mass shift) between the tools, all of which likely contribute substantially to the differences in number of PSMs obtained.
- o The analysis presented in Figure 8 requires revision. The union of multiple analyses of the same data with individual 1% FDR rates does not itself have a 1% FDR. True positives are more likely than false positives to generate high scores and thus are more likely to pass a given confidence threshold, making a given true positive more likely than a false positive to be accepted in each individual analysis of the data (and thus to be in common between them). When the union of these individual analyses is taken, the true positives are thus more likely to be redundant than false positives, leading to an effective enrichment of false positives in the union, increasing the FDR above 1% (possibly very far above 1%, depending on the analyses in question). As such, the "any FDR <1%" category, a union of 3 result sets, does not maintain the 1% FDR of its component sets. It should not be compared to other analyses that use a 1% FDR and should be removed from Fig. 8.
- o The authors state that they are able to obtain a similar number of PSMs as the MSFragger analysis when considering their union of PSMs passing any of FDRs, or over 60,000 PSMs when considering the union and applying glycan network smoothing. However, as noted above, the union is no longer at 1% FDR and is not comparable to an analysis that is at 1% FDR. These statements should be removed.
- o Additional discussion of the peptide-only FDR result is needed. The authors' contention that the additional PSMs obtained by MSFragger can be attributable to the different FDR is undercut by the fact that the peptide-only FDR is the most similar to MSFragger's method, but is by far the most conservative metric in Figure 8, obtaining far fewer PSMs than the joint FDR method. How was this peptide-only FDR calculated?
- o The authors state that the "bin-specific probabilities in PeptideProphet may be learned from a small number of observations" for a given mass shift, which is not correct. The mass model used in PeptideProphet models the probability as a result of the number and score of PSMs in each mass bin relative to the number and score of PSMs in other bins, as well as the proportion of targets and decoys in that bin. In a mass bin with few PSMs, the resultant probability is primarily a result of the low number of PSMs relative to other bins, meaning the mass bin probabilities are modeling using information from the complete dataset even if there are only a few PSMs in a given bin. This paragraph should be revised to reflect this.

The comparison with MSFragger has been completely removed. The reviewer is correct that our comparison was flawed and based upon faulty assumptions.

Some figures need graphical revisions: Table 2: The features used to predict spectra are not clearly defined, and the mass of them presented in the table (while somewhat visually impressive) does not add much to understanding. Terms used should be defined either in the table or supporting information (e.g. what is stub glycopeptide?). The table could also be greatly simplified by combining some iterative entries (e.g. glycan loss 0-20 instead of listing glycan loss 0, glycan loss 1, etc.).

Thank you for the recommendation, this table has been moved to the supplement and specific terms have been defined.

Figure 6: The goal of the figure is to show similarity between observed and predicted spectra, but the visual display of the data makes this very hard to assess. All spectra need to be increased in size, especially in the vertical dimension to aid in distinguishing the heights of peaks (perhaps tile all 4 vertically rather than a 2x2 grid)

Thank you. This has been addressed, with predicted spectra figures moved to the supplement where sufficient space was made available.

Overlapping text labels for peaks need to be moved so they can be read

Thank you for the suggestion, we have removed the tightly clustering oxonium ion peak annotations in the predicted spectra to make the less densely packed peak annotations visible.

Text labels need to be moved far enough away from peaks that peak height can be accurately distinguished. All text needs to be increased in size to be visible without requiring excessive zooming. If this is not feasible for all individual peaks, perhaps a visual representation of the ion types could be substituted for long text labels.

Thank you for the suggestion. We have increased the size of all annotated spectra and reduced the label cluttering.

Reviewer #3 (Remarks to the Author):

The manuscript by Klein et al., is an impressive update to the GlycReSoft glycoproteomic bioinformatics pipeline from the Zaia research group. The manuscript describes an informatic approach which increases the assignment of glycopeptides in complex MS datasets by cross-referencing the measured values of this dataset with an in-silico search space that is guided by a hybridisation/combination of several algorithms such as a glycopeptide spectral matching algorithm, a glycan coverage approximation, inter-peak relationships, peak intensity predictions and a glycome network smoothing. Whilst much of these approaches have been reported in previous publications, several improvements to approaches are discussed and presented in comprehensive set of scoring algorithms. The combination of approaches are well thought out in-silico approach that increases the number of confident assignments whilst keeping the FDR low. The statistics do support the improvements.

Whilst the paper is impressive, it is very heavy on the algorithms and mathematical approach and subsequent validation with previously generated datasets. As such, the findings here may be lost on the audience who would most benefit from this - glyco-analytical scientists. The reviewer has therefore some queries and guiding comments with the intention to helping the paper to be more convincing to an audience who would most benefit from this manuscript and yet may be overwhelmed by the informatic approach;

is the calculated FDR the only measure of accuracy we have for this dataset? how are we certain the improvements to the assignments are real outside of your informatic approach? would it be feasible to demonstrate your approach with additional experiments with exoglycosidases to demonstrate the efficacy of your assignments?

Thank you for the suggestion. All of the work done here is based upon publicly available datasets. The only comparably sized dataset which included exoglycosidase treatment, PXD025859 from the StrucGP original publication, was acquired using high and low energy HCD settings in separate scans, which wouldn't be consistent with the sceHCD datasets we've shown here to validate the fragmentation model.

We now include a retention time model that is independent of the scoring function, and controls for wild

deviations in glycan composition. To see the model coefficients and the deviations, please see supplementary Section 9 for the base searches, and see supplementary Figures 9-11 in supplementary Section 7 for the overlap between the different scoring functions, post-retention time updates.

can you comment on what extra glycopeptides you were able to identify from the comparison of the base model with the glycan network smoothing fragmentation model on these datasets?

Thank you for the recommendation. We provide an example in Figure 2j showing an MS2 spectrum that was rescued by network smoothing, and further expand on it in the supplementary Section 8 and Figure S12 which depicts a set of rescued glycoforms for the same glycosylation site on the MS1 level. The rescued glycoforms are consistent with the retention time model learned for the sample in the base search, which provides a form of internal validation, though it is not completely orthogonal.

the dataset which you used was generated using an orbitrap fusion with stepped collision in HCD. whilst this is an ideal situation, there does remain the question of how well your approach would be to other types of mass spectrometry formats. For example, can you demonstrate any gains or losses of your approach to time of flight mass spectrometers? Would the beam type formats of mass spectrometers would affect the peak intensity prediction?

We show in Section 2.3 that our intensity prediction model suffers greatly when data acquisition conditions deviate from those found in the training data, even when it is the same instrument model used at different collision energies, which is consistent with other models of this kind. If we made collision energy a regression parameter and incorporated data from multiple sources, we may be able to overcome this, but few sceHCD studies have been published, and it would be difficult to mix them with single-energy multiple-scan orbitrap datasets like PXD025859. We are not aware of any large-scale time-of-flight glycoproteomics datasets of comparable size, although there is no reason why the method we use here would not work.

can you explain why you didnât consider the oxonium ions into your scoring method when in fact, these ions are very valuable diagnostic in glycopeptide identification?

Oxonium ions are indeed very important, but we did not try to model them because they are difficult to deconvolve in the presence of co-isolation. We observed that simpler rule-based methods worked acceptably, without concerns that coisolation corrupted the signature outright. We include signature ions in all of our scoring models as well, see Eqs 6-8 in Section 5.2. Implicitly, the glycan composition revision rules described in supplementary Section 1.1.2 page 4 draw on the same assumption that oxonium ions being abundant cannot easily be ârevised awayâ. Unfortunately, this does have the ability to âlock inâ incorrect glycan compositions when co-isolation is rampant, though we view this as no worse than other methods.

some of the figures such as figures 4, 5, 6 are incredibly hard to read and are of very low resolution. Since they are an integral part of your thesis, could you please provide higher resolution images for this manuscript

Thank you, we have recreated all figures in the rewrite with this in mind.

Version 1:

Reviewer comments:

Reviewer #1

(Remarks to the Author)

This paper was almost re-written and most of my concerns have been addressed. The software seemed to be much improved as well by considering more features such as RT prediction. It is also good to see that the package is rewritten in Python 3, meaning that the package can be either used as a standalone program or a Python library.

There are still a few new issues in the revised manuscript:

1. In Fig.1c, it introduced "glycan index" but it is not clear to me about the methods, especially when GlycReSoft only used glycan compositions. What glycan fragments does GlycReSoft use? And how to index them?
2. $\sigma = 5e-6$ in eq. 8. And as ppm is 10 or 20, $(\text{ppm}-u)^2/\sigma$ is always a very large value (eq. 3). Does eq. 3 work as expected? Does ppm refers to Da value converted from ppm tolerance?
3. Fig 2j is too hard to read due to too many overlap annotations over peaks. "isolation purity" is not defined, why the value is

zero? I suggest to use simple annotations like N/H/F/A instead of full glyco names to plot the texts.

4. Fig 2c, RT in seconds or minutes?

5. In the glycan connectivity graph, it is not clear that if RT shifts of some glyco units (e.g., NeuAc) are considered. If not, expanding from HexNAc/Hex/Fuc to NeuAc using glycan network may result in false mapping. Is MS1 information displayed in Fig. 1f considered in the glycan network?

6. Although ID numbers are compared with pGlyco3, but search speed was not considered in the comparisons.

7. A minor issue is that if users would like to use GlycReSoft as a Python library, the documentation is not complete enough.

Reviewer #2

(Remarks to the Author)

The revised manuscript from Klein et al is nearly a complete rewrite, incorporating a new retention time-based composition correction algorithm into the described workflow and reanalyzing all data and improving all figures. The revision addresses nearly all my concerns from the original manuscript and is a major improvement. While the results do not show dramatic improvements compared to existing software, the idea of using glycopeptide retention time and spectrum prediction alongside biosynthetic network smoothing for glycopeptide identification is excellent and clearly described in the manuscript.

Specific comments:

- In the comparison with pGlyco3, different glycan databases were used for GlycReSoft and pGlyco3, meaning the number of GPSMs found by each method is a function of both the underlying algorithms and the glycan databases. This caveat must be mentioned in the main text when referring to the results in Figure 3, along with the sizes of the glycan databases used and any key differences (if there are any).
- In the yeast entrapment analysis, what was the yeast glycome reference used and how many (and which) glycans did it contain? 3.3% is a pretty high entrapment rate given an expected FDR of 1%, especially given the simplicity of the yeast glycome. But if the size of the entrapment glycome is much larger than that of the yeast it could explain the high entrapment rate. It would also be nice to see if there were any common characteristics of the entrapment glycans matched (e.g., fucose-containing vs not) in the supporting information.
- The use of retention time to correct mis-identified isobaric glycans is a great idea, but I didn't see how many such corrections were able to be made in any of the analyses. Figure 2 shows the increase in GPSMs from the base model to with fragmentation and retention time modeling, but presumably some of that increase is from the fragmentation modeling. Would removing incorrect glycans via retention time modeling also allow for an increase in GPSMs from removing some previously identified decoys?
- (minor) in section 5.1.3, a sentence is missing the second half: "We followed the mass accuracy settings suggested in each dataset's original publication, but found that the number and type of adducts to consider were ."

Reviewer #4

(Remarks to the Author)

I agree with the previous reviewers that the paper is well written and was a nice summary of extensive works on the GlycReSoft software. The revision has adequately addressed the concerns raised by the reviewers in the previous round. I have a few minor suggestions that the authors may consider in their final version of the manuscript.

1. The scoring function is empirical but quite complex. It is useful to provide some intuition how the forms and parameters are chosen and how each component of the scoring function actually benefit the glycopeptide identification. Such discussion can be helpful to understand if how the approach can be extended with the evolving tandem mass spectrometry technology (e.g., with various instrumentations and fragmentation methods).

2. The authors used a unique approach to estimate FDR in glycopeptide identification. It is not completely clear how accurate this method is (in fact, not standard method exists for FDR estimation in glycopeptide identification). I suggest the authors consider an independent testings, i.e., to compute the similarity the spectra of identified as the same glycopeptide, where the outlier spectra are likely misidentified.

3. Some formulas referred to is hard to be matched. E.g., Eq. S12 referred to in Section 2.2.2 (pg 5).

Author Rebuttal letter:

REVIEWER COMMENTS

Reviewer #1 (Remarks to the Author):

This paper was almost re-written and most of my concerns have been addressed. The software seemed to be much improved as well by considering more features such as RT prediction. It is also good to see that the package is rewritten in Python 3, meaning that the package can be either used as a standalone program or a Python library.

There are still a few new issues in the revised manuscript:

1. In Fig. 1c, it introduced glycan index but it is not clear to me about the methods, especially when GlycReSoft only used glycan compositions. What glycan fragments does GlycReSoft use? And how to index them?

Thank you, this is a good question. The glycan compositions fragments are generated using a glycan type specific procedure that makes assumptions about the core motif. The N-glycan-specific process is described in pseudocode in supplementary Listing 1. The indexing strategy is otherwise implemented based upon the pGlyco3 glycan index.

2. $\sigma = 5e-6$ in eq. 8. And as ppm is 10 or 20, $(\text{ppm}-u)^2/\sigma$ is always a very large value (eq. 3). Does eq. 3 work as expected? Does ppm refers to Da value converted from ppm tolerance?

Thank you for point this out. The value used here refers to the difference in raw parts, not parts-per-million. We have rewritten this in terms of parts-per-million, so instead of $5e-6$, this is 5 PPM.

3. Fig 2j is too hard to read due to too many overlap annotations over peaks. Isolation purity is not defined, why the value is zero? I suggest to use simple annotations like N/H/F/A instead of full glyco names to plot the texts.

This figure has been redrawn using SNFG glyphs to indicate monosaccharide compositions on fragments. The isolation purity label was described in a previous publication, indicating the isolation window for the precursor ion is contaminated by other peaks. In this case the deconvolution algorithm detects five overlapping isotopic patterns which exceed the signal of the selected ion and in one case overlap it.

4. Fig 2c, RT in seconds or minutes?

The unit of retention time used throughout the article is minutes as in Fig 2d, and we have also updated the figure caption to reflect this.

5. In the glycan connectivity graph, it is not clear that if RT shifts of some glyco units (e.g., NeuAc) are considered. If not, expanding from HexNAc/Hex/Fuc to NeuAc using glycan network may result in false mapping. Is MS1 information displayed in Fig. 1f considered in the glycan network?

If the concern is that Fig 1k does not include a sialic acid, this was not drawn due to limited space. The glycan network smoothing model does include sialic acids, their presence determines which neighborhoods a glycan maps to, as in Fig 2i there are sialo neighborhoods which exclude sialylated compositions. After MS2 identification, regardless of search strategy, chromatographic feature consensus (Fig 1f), adduct deconvolution (Fig 1g), and retention time modeling (Fig 1h-i) are used to refine the assigned glycan composition on that peptide sequence. A note to emphasize this has been added to the text in Section 5.1.3.

6. Although ID numbers are compared with pGlyco3, but search speed was not considered in the comparisons.

GlycReSoft is much slower than pGlyco3, from the perspective of the hardware. While it runs in comparable amount of time to pGlyco3 on a single sample, it does so by consuming substantially more resources. This is broken down in supplementary Section 14.

7. A minor issue is that if users would like to use GlycReSoft as a Python library, the documentation is not complete enough.

Making GlycReSoft a fully functional library was not a goal, and much of the plumbing logic is impractical to use programmatically. If new features are added, more work will be made on documentation.

Reviewer #2 (Remarks to the Author):

The revised manuscript from Klein et al is nearly a complete rewrite, incorporating a new retention time-based composition correction algorithm into the described workflow and reanalyzing all data and improving all figures. The revision addresses nearly all my concerns from the original manuscript and is a major improvement. While the results do not show dramatic improvements compared to existing software, the idea of using glycopeptide retention time and spectrum prediction alongside biosynthetic network smoothing for glycopeptide identification is excellent and clearly described in the manuscript.

In the comparison with pGlyco3, different glycan databases were used for GlycReSoft and pGlyco3, meaning the number of GPSMs found by each method is a function of both the underlying algorithms and the glycan databases. This caveat must be mentioned in the main text when referring to the results in Figure 3, along with the sizes of the glycan databases used and any key differences (if there are any).

Thank you for the suggestion. This is now shown in supplementary figure 12, and referenced early in the methods section to highlight that the databases are different. There is also a note that the distinction between databases does not change markedly between the mouse tissue dataset and the yeast entrapment dataset.

â€ In the yeast entrapment analysis, what was the yeast glycome reference used and how many (and which) glycans did it contain? 3.3% is a pretty high entrapment rate given an expected FDR of 1%, especially given the simplicity of the yeast glycome. But if the size of the entrapment glycome is much larger than that of the yeast it could explain the high entrapment rate. It would also be nice to see if there were any common characteristics of the entrapment glycans matched (e.g., fucose-containing vs not) in the supporting information.

We initially thought this was due to ammonium adduction and fucosylation as well, but in reality it was something far more subtle having to do with how peptide mass filtering differs between GlycReSoft and pGlyco3. The details of this are unpacked in supplementary section 10, which includes tables enumerating the contributing mouse glycan and mouse peptides.

â€ The use of retention time to correct mis-identified isobaric glycans is a great idea, but I didn't see how many such corrections were able to be made in any of the analyses. Figure 2 shows the increase in GPSMs from the base model to with fragmentation and retention time modeling, but presumably some of that increase is from the fragmentation modeling. Would removing incorrect glycans via retention time modeling also allow for an increase in GPSMs from removing some previously identified decoys?

Accurately tracking the number of revisions that occur is difficult because revisions occur at the chromatographic peak level, and the same identification may be updated many times. The terminal state of a different from starting is knowable, and varies from sample to sample. We have added a new supplementary section 15 to the supplementary material listing the number of revisions and approximate percentages. These tables show that between ~1-6% of glycopeptide features were revised in the mouse tissue dataset, and 4-6% of glycopeptide features in the human serum dataset.

â€ (minor) in section 5.1.3, a sentence is missing the second half: â€We followed the mass accuracy settings suggested in each dataset's original publication, but found that the number and type of adducts to consider were .â€

Thank you for catching this, a thought was interrupted. The broken line has been fixed.

Reviewer #4 (Remarks to the Author):

I agree with the previous reviewers that the paper is well written and was a nice summary of extensive works on the GlycReSoft software. The revision has adequately addressed the concerns raised by the reviewers in the previous round. I have a few minor suggestions that the authors may consider in their final version of the manuscript.

1. The scoring function is empirical but quite complex. It is useful to provide some intuition how the forms and parameters are chosen and how each component of the scoring function actually benefit the glycopeptide identification. Such discussion can be helpful to understand if how the approach can be extended with the evolving tandem mass spectrometry technology (e.g., with various instrumentations and fragmentation methods).

Thank you for the suggestion. We attempted to elaborate on the reasoning behind our scoring function's construction in the text, and color-coded parts of their equations to their justification in prose. Additionally, we added some text describing certain assumed conditions, which if no longer satisfied would suggest a revision of how information is combined, especially if the model could be improved.

2. The authors used a unique approach to estimate FDR in glycopeptide identification. It is not completely clear how accurate this method is (in fact, not standard method exists for FDR estimation in glycopeptide identification). I suggest the authors consider an independent testings, i.e., to compute the similarity the spectra of identified as the same glycopeptide, where the outlier spectra are likely misidentified.

Thank you for the suggestion. Our approach to FDR estimation is identical to pGlyco3's process, with the exception that we replace the traditional target-decoy analysis of the peptide score with a linear SVM of peptide-dependent features. Our glycan FDR approach, as well as the process by which the peptide and glycan FDRs are joined are identical to the process described in the original pGlyco2 article. The linear SVM is itself a simplified adaptation of the process used in Percolator or MokaPot where several features are combined.

We added a supplementary table 6 in supplementary section 11 which shows that the the average intra-cluster similarity did not change appreciably with the addition of successive modeling methods.

3. Some formulas referred to is hard to be matched. E.g., Eq. S12 referred to in Section

2.2.2 (pg 5).

Unfortunately, the retention time modeling component involves some lengthy calculations and they together occupy more space than the rest of our methods combined, and as such had to be moved to the supplementary material. As useful as the retention time model is, it is an extension of previously published work that serves to stabilize the identifications as a source of training data for the other post-search model fitting steps.
